# Clusiomitidae, A New Family of Eocene Fossil Acalyptratae, with Revision of *Acartophthalmites* Hennig and *Clusiomites* Gen. Nov. (Diptera) [note 1]

**DOI:** 10.3390/insects12121123

**Published:** 2021-12-15

**Authors:** Jindřich Roháček, Christel Hoffeins

**Affiliations:** 1Entomology, Silesian Museum, Nádražní Okruh 31, 746 01 Opava, Czech Republic; 2Liseistieg 10, D-22149 Hamburg, Germany; chw.hoffeins@googlemail.com

**Keywords:** extinct acalyptrate flies, ancient diversity, amber forest, Paleogene

## Abstract

**Simple Summary:**

This study contributes to knowledge of the diversity of tertiary fossil flies of the group Acalyptratae. A new family Clusiomitidae is described on the basis of seven (four new) species from Baltic amber inclusions (Eocene, ca 48–34 Ma) belonging to two genera, *Clusiomites* gen. nov. and *Acartophthalmites* Hennig, 1965. Discovery of a new acalyptrate family in Baltic amber indicates an unusually rich diversity of this group in the so called “amber forest” covering a vast area of Europe in the Eocene climatic optimum.

**Abstract:**

The Eocene Baltic amber fossil flies of the genus *Acartophthalmites* Hennig, 1965 (Diptera: Acalyptratae) are revised. Seven species are recognized and described or redescribed. Five species, *A. crassipes* sp. nov., *A. luridus* sp. nov., *A. rugosus* sp. nov., *A. tertiaria* Hennig, 1965 (type species) and *A. willii* Pérez-de la Fuente, Hoffeins et Roháček, 2018 are retained in *Acartophthalmites* while *Clusiomites* gen. nov. is described for two other species, *C. clusioides* (Roháček, 2016) comb. nov. (type species) and *C. ornatus* sp. nov. Relationships of these fossil taxa are discussed and, because they cannot be confidently placed in any known family of Diptera, a new family, Clusiomitidae, is established for them. Clusiomitidae is recognized as a family of Opomyzoidea, probably most closely allied to Clusiidae. These results again confirmed that the diversity of acalyptrate flies was very high in the Mid-late Eocene amber forest.

## 1. Introduction

Acalyptratae is a relatively young paraphyletic taxon of two-winged flies (Diptera) belonging to Schizophora. The oldest fossil records of Acalyptratae are from the Eocene Baltic amber, cf. [1,2,3,4,5,6]. Significantly and surprisingly, the acalyptrate flies were already unexpectedly diverse in this amber period (Mid-late Eocene, 48–34 Ma, see [7,8] and habitat (the so called Baltic amber forest, see [9,10,11,12]). The majority of extant families of Acalyptratae were well-represented in the Baltic amber in addition to members of several extinct families only known from these amber fossils [13]. Two additional ancient acalyptrate families, the Hoffeinsmyiidae Michelsen, 2008 and Yantaromyiidae Barták, 2019, have recently been described from Baltic amber [14,15]; however, there are some further extinct genera and species of which the family affiliation remains uncertain. As demonstrated by a revisionary study of fossil Anthomyzidae [16] and its subsequent additions [17,18,19], the species diversity of some acalyptrate families was also even richer in the Eocene amber forest than today in the whole of Europe.

The genus *Acartophthalmites* Hennig, 1965 is one of these unclassified and somewhat mysterious Baltic amber taxa. It was originally described by Hennig [1] as a monotypic genus for *A. tertiaria* Hennig, 1965 and classified as a tentative member of the extant family Acartophthalmidae. Roháček [20], when describing a second species of *Acartophthalmites*, discussed the family affiliation of this genus, demonstrated that it cannot belong to Acartophthalmidae and suggested the druid flies (Clusiidae) as its most probable relatives. This affinity has also subsequently been supported by Pérez-de la Fuente et al. [21] on the basis of study of a third species, *A. willii* Pérez-de la Fuente, Hoffeins & Roháček, 1918. In both latter studies, it has been proposed to revise all available specimens of *Acartophthalmites* to obtain as complete as possible morphological data for more precise consideration of its relationships and taxonomic classification.

The present study is therefore aimed at revision of the genus *Acartophthalmites* and its allies. Although not all specimens obtained for examination were suitable for description (due to poor state of preservation), we gained a good deal of new morphological information which enabled us to (1) recognize seven species including four new, (2) redefine and redescribe the genus *Acartophthalmites*, (3) establish a new genus, and (4) describe a new fossil family to include all these taxa.

## 2. Materials and Methods

### 2.1. Material

A total of 26 amber samples with *Acartophthalmites*-like inclusions have been examined. They are deposited in CCHH–collection of C. and H. W. Hoffeins, Hamburg, Germany, CFKH–collection of Friedrich Kernegger, Hamburg, Germany; CMTB–collection of M. von Tschirnhaus, Bielefeld, Germany, GPIH–Geological-Palaeontological Institute and Museum, University of Hamburg, Germany; GZG–Geoscience Centre and Museum, University of Göttingen, Germany; MCZC–Museum of Comparative Zoology, Harvard 118 University, Cambridge, Massachusetts, USA. Note: the type of specimens from CCHH will subsequently be deposited in the Senckenberg Deutsches Entomologisches Institut, Müncheberg, Germany (SDEI).

### 2.2. Preparation of Amber Specimens

The methods of preparation of amber stones with fly inclusions are described in detail in [13]. The amber specimens examined had already been cut, ground and polished as close and as parallel as possible to the frontal, dorsal and lateral sides of the fly and one of them had been subsequently embedded in artificial resin (also ground and polished) [22] in order to facilitate its stereoscopic investigation.

### 2.3. Techniques of Investigation

The amber inclusions have been examined, drawn and measured using two types of binocular stereoscopic microscopes (Reichert, Olympus). Legs were drawn on squared paper using a Reichert binocular microscope with an ocular screen. The specimens were either photographed by digital camera Canon EOS 60D with macro lens Canon MP-E 65 mm 1–5× or by digital camera Canon EOS 5D Mark III with a Nikon CFI Plan 10×/0.25NA 10.5 mm WD objective attached to Canon EF 70–200 mm f/4L USM zoom lens. The specimen photographed by means of the latter equipment was repositioned upwards between each exposure using a Cognisys StackShot Macro Rail and the final photograph was compiled from multiple layers (35) using Helicon Focus Pro 7.0.2. The final images were edited in Adobe Photoshop CS6. Some illustrations were drawn using the obtained macrophotographs in which details were inked based on direct observation at higher magnification using a binocular microscope. Measurements: Six characteristics were measured–body length (measured from anterior margin of head to end of cercus, thus excluding the antenna), wing length (from wing base to wing tip), wing width (maximum width), index Cs_3_:Cs_4_ (=ratio of length of 3rd costal sector:length of 4th costal sector), index r-m\dm-cu:dm-cu (=ratio of length of section between r-m and dm-cu on discal cell:length of dm-cu) and index r-m\dm-cu:CuA_1_ (=ratio of length of section between r-m and dm-cu on discal cell:length of apical portion of CuA_1_). 

### 2.4. Morphological Terminology

Follows that used in [20,21], including terms of the male hypopygium to be in continuation with [16] on fossil Anthomyzidae. Male terminalia terminology is largely based on the “hinge” hypothesis of the origin of the eremoneuran hypopygium, re-discovered and documented by [23] and, therefore, the following alterations of terms of the male genitalia (against those used by other hypotheses) need to be listed (term used here first): epandrium = periandrium, gonostylus = surstylus. Morphological terms of the male abdomen and terminalia are depicted in Figure 4A,B, Figure 6A and Figure 21A,B; those of the female abdomen in Figure 8A,B, Figure 18A–C and Figure 20C. The synonymous morphological terms of adult structures and their abbreviations as used in the most recent manual of Afrotropical Diptera [24] are given in parentheses in the list of abbreviations below.

A_1_—anal veinac—acrostichal (setulae)ar—aristaC—costace—cercusCs_2_, Cs_3_, Cs_4_—2nd, 3rd, 4th costal sectorCuA_1_—cubituscx_1_, cx_2_, cx_3_—fore, mid, hind coxadc—dorsocentral setaedm—discal medial celldm-cu—discal medial-cubital (=posterior, tp) cross-veinep—epandriumf_1_, f_2_, f_3_—fore, mid, hind femurfrt—frontal trianglegs—gonostylusha—haltere hu—humeral (=postpronotal) (seta)hum—humeral cross-veinlgs—left gonostylusM—mediamspl—mesopleural (=anepisternal) (seta)npl—notopleural (seta)oc—ocellar (seta)ors—fronto-orbital (seta)pa—postalar (seta)pk—preapical kink on R1poc—postocular setulaeppl—propleural (=proepisternal) (seta)prs—presutural (seta)pvt—postvertical (seta)R_1_—1st branch of radiusR_2+3_—2nd branch of radiusR_4+5_—3rd branch of radiusrgs—right gonostylusr-m—radial-medial (=anterior, ta) cross-veinS1–S10—abdominal sternasa—supraalar (seta)sc—scutellar (seta)Sc—subcostasctl—scutellum stpl—sternopleural (=katepisternal) (seta)T1–T10—abdominal tergaT6 + S8–dorsal pregenital synscleritet_1_, t_2_, t_3_—fore, mid, hind tibiatr_1_, tr_2_, tr_3_—fore, mid, hind trochantervi—vibrissavte—outer vertical (seta)vti—inner vertical (seta)

## 3. Results

### 3.1. Systematic Palaeontology

Class Insecta Linnaeus, 1758.Order Diptera Linnaeus, 1758.Superfamily Opomyzoidea Fallén, 1820.

#### 3.1.1. Family *Clusiomitidae* fam. nov

LSID urn:lsid:zoobank.org:act:36B10CD1-C037-45FC-B70A-E07105C1C5BD

Type genus: *Clusiomites* gen. nov., designated here.

**Diagnosis.** Body slender to relatively robust, length approximately 3–5 mm (Figure 1A,B). Antenna porrect; arista dorsobasal; pedicel with distal margin of outer side simple, at most slighly convex. Cephalic setae strong; pvt relatively long and robust, divergent and closely inserted; vti by far longest cephalic seta, usually slightly exclinate to reclinate; oc divergent and usually smaller than pvt; three reclinate ors; no interfrontal setae; vibrissa small but distinct, strongly medially curved. Thorax with two or three dc; two pa, anterior very long (usually longest thoracic seta); two sc; one long mspl and a number of setulae on mesopleuron; one long (posterior) stpl besides small setulae; one long, upcuved ppl; mesopleuron with posterior dorsal ridge reduced; prosternum entirely bare; t_1_ and t_2_ with small dorsal preapical setae; t_2_ with two or three short dorsal setae distally (most distal one representing dorsal preapical seta) and with a long row of erect posterior to posterodorsal setae; f_2_ markedly prolonged and often thickened, as long as or longer than f_3_; cx_3_ ventrally with distinct setose process projecting posteroventrally. Wing clear or patterned, often anterodistally or entirely infuscated; C complete, without breaks; Sc complete and ending in C far from apex of R_1_; R_1_ with distinct kink at level of end of Sc, bare or setulose; alula broad and relatively large. Male: postabdomen with large and long dorsal synsclerite (probably a fusion of T6 and S8 and therefore termed T6 + S8); T7 absent; S6 and S7 obviously asymmetrical and situated left ventrolaterally to laterally; epandrium short and high, down-directed and open ventrally, and with anal fissure directed ventrally; male cerci separate, free, inserted below anal fissure; gonostyli slightly to distinctly asymmetrical and sometimes bilobed. Female: preabdomen broad, with transverse T1–T5; S2–S5 markedly narrower so that pleural membrane large; postabdomen rapidly tapered posteriorly but 6th and 7th segments usually with sclerites more or less transverse; 8th and 10th segment markedly narrower; cerci slender, elongate and setose.

**Figure 1 insects-12-01123-f001:**
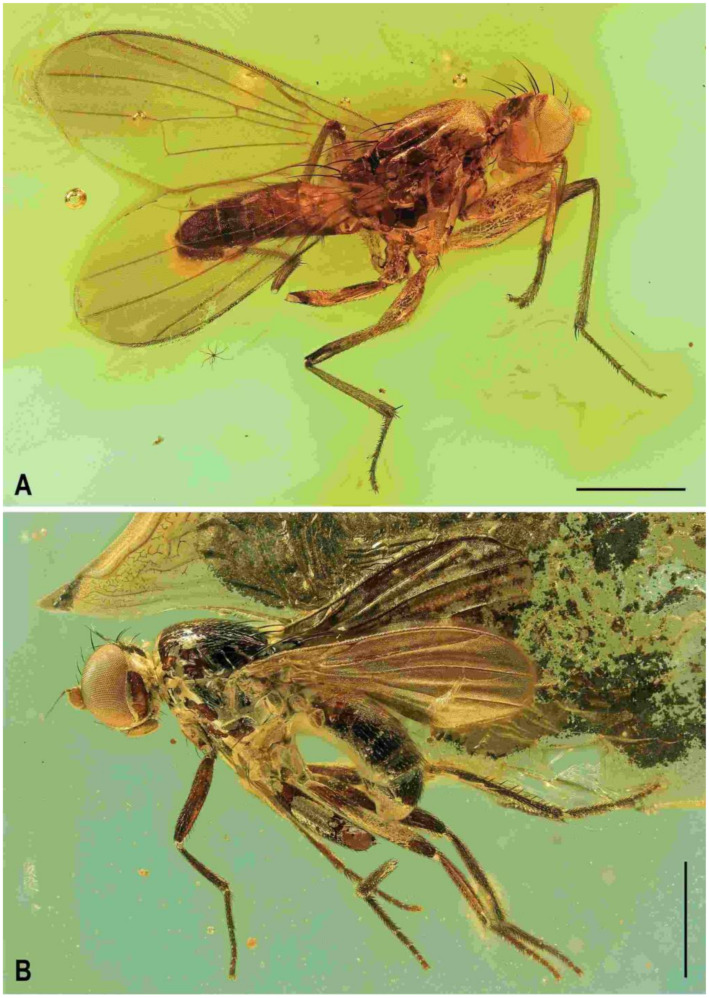
Representatives of Clusiomitidae: (**A**) *Clusiomites clusioides* (Roháček), male holotype, right laterally; (**B**) *Acartophthalmites crassipes* sp. nov., male holotype, left laterally. Scales: 1 mm. Photos by J. Roháček.

**Definition.** Adult. ***Head*** as long as high (*Clusiomites*) or higher than long (*Acartophthalmites*). Occiput dorsally concave; frons relatively narrow; frontal lunule present; face flat to slightly convex, with shallow concavities below antennae; eye large, strongly convex and bare; gena very low; postgena expanded due to posteroventral excavation of eye; palpus small and slender; antenna porrect and relatively small; pedicel with both outer and inner distal margins simple, without extension or process; first flagellomere rounded and laterally flattened; arista dorsobasal with shortly cylindrical basal segment and its terminal part with short to relatively long ciliation. Chaetotaxy: pvt (strong, divergent and closely inserted); vti (longest cephalic seta), vte (strongly exclinate); oc (inserted between ocelli, shorter than pvt); three ors (all reclinate); no interfrontal setae but a few microsetulae on frons sometimes present; postocular setulae numerous, dorsally in more than one row; a number of additional erect setulae on adjacent lateral parts of occiput and postgena; postgena with two or three posteroventral setae in addition; occiput with a brush of small setulae above foramen; vi distinct but short, curved medially; subvibrissa not developed; peristomal setulae small and sparse.

***Thorax*** slightly narrower than head. Humeral callus (postpronotum) well developed; scutum with transverse suture present only laterally; scutellum subtriangular with rounded apex; subscutellum small but distinct; mesopleuron with posterior dorsal ridge reduced. Chaetotaxy: one hu, two npl, one sa, two pa (anterior very long and strong, usually longest thoracic seta), no prs; two or three dc (all postsutural); scutum covered by relatively dense microsetae (shorter and denser anteriorly); ac microsetae arranged in about 8–12 rows in front of suture; 1 prescutellar ac (short to relatively long); two sc (apical strong, laterobasal shorter but relatively robust); one ppl (upcurved and relatively long); one mspl and numerous microsetae on mesopleuron; one stpl and a number of microsetae sternopleuron; prosternum always bare. 

***Legs***. Fore, mid and hind legs strikingly differing in length of their femora and tibiae; mid leg distinctly to strikingly prolonged, f_2_ as long as or longer than than f_3_; cx_3_ with strong ventral process; f_1_ with a short row of longer posteroventral setae on distal third and a few posterodorsal setae in the middle third of femur; f_2_ simply setulose or with specialized ventral setosity; t_1_ with one or two small dorsopreapical setae; t_2_ with a long row of posterior erect setae, two or three short dorsal setae (including one preapical) and one longer and thicker ventroapical seta; t_3_ without dorsopreapical seta, but with one longer ventroapical and one shorter anteroapical seta; tarsi simple, slender; mid and hind basitarsi long (as long as rest of tarsus); claws relatively small.

***Wing*** relatively broad to distinctly elongate; membrane more or less brownish infuscated, often in anterodistal half. C extended beyond apex of R_4+5_ where attenuated and this slender nebulous part usually reaching to M; C finely setulose but basally with a pair of longer setae and Cs_2_ (sector between apices of R_1_ and R_2+3_) with thicker sparse spine-like setulae in addition. No costal break. Sc fine but complete, ending into C far from apex of R_1_. R_1_ short, robust, bare or setulose, having a distinct preapical kink (see Figure 3H, pk) on the level of end of Sc. R_2+3_ long but ending distinctly farther from wing apex than M. R_4+5_ more or less bent posteriorly, ending near apex of wing; discal (dm) cell relatively elongate; anterior cross-vein (r-m) situated near middle of cell; distal part of CuA_1_ relatively short and reaching or almost reaching) wing margin; A_1_ long (more rarely short) but ending far from wing margin. Cells bm and cup closed. Anal lobe well developed. Alula small to relatively large, broad. 

***Male abdomen*** slender to moderately broad, elongate, usually pyriform in dorsal view, with apex down-curved due to large dorsal pregenital sclerite; preabdominal terga transverse, all distinctly setose; T1 and T2 separated only laterally, dorsally coalesced. Preabdominal sterna S2–S5 finely setulose, much narrower than associated terga (pleural membrane large) and usually becoming slightly wider posteriorly, with S5 largest and broadest. S1 smallest, short and bare. Postabdomen with sclerites well developed, dark-pigmented and asymmetrical. T6 not present as separate sclerite but probably completely fused with S8 to form with it a large, long and slightly asymmetrical dorsal synsclerite termed here T6 + S8. S6 and S7 asymmetrical, situated left laterally to lateroventrally (S6). 

***Male genitalia***. Epandrium short and high, with posterior side oriented ventrally due to enlarged synsclerite T6 + S8. Anal fissure high and dorsally narrowed, ventrally widened and hence epandrium ventrally open. Gonostyli relatively small, asymmetrical and bilobed in *Clusiomites*, probably simple and more symmetrical in *Acartophthalmites*. Cerci also asymmetrical in *Clusiomites* but symmetrical in *Acartophthalmites* and both free (separate) and situated ventrally below anal fissure. Internal genitalia (aedeagal and/or hypandrial complex) unknown. 

***Female abdomen*** narrow and elongate in *Clusiomites*, broad in *Acartophthalmites*, subovoid in dorsal view, gradually tapered posteriorly. Preabdominal terga shorter and more transverse than in male, all setose. Preabdominal sterna shortly and finely setose, all relatively narrow (slightly narrower than in male). S1 probably small and short (as in male); S2–S5 becoming somewhat wider posteriorly, S5 widest.

***Female postabdomen*** strongly tapered posteriorly, with 7th to 10th segments retractable. T6 much smaller than T5, narrower, shorter and distinctly transverse; T7 slightly longer and distinctly narrower than T6, both sparsely setose. T8 probably small and narrow; T10 very small, short, obviously with a single pair of setae. S6 shorter and wider than S5, transverse; S7 usually longer and narrower than S6; S8 small and short, sometimes divided into two parts; S10 small, semicircular. Cerci slender, elongate, with rich setosity, with apical and lateral setae longest.

Preimaginal stages. Unknown, except for egg–described below under the genus *Acartophthalmites*.

**Genera included**: *Clusiomites* gen. nov., *Acartophthalmites* Hennig, 1965.

#### 3.1.2. Genus *Clusiomites* gen. nov

LSID urn:lsid:zoobank.org:act:D05E63E8-ACCA-4EE4-BD61-B51CE57B30FA

Type species: *Acartophthalmites clusioides* Roháček, 2016, designated here.

**Etymology**: The name of the new genus is an abbreviated conjunction of *Clusio* [des + Acartophthal] *mites*, gender masculine.

**Diagnosis.** Body slender, somewhat resembling those of *Clusiodes* species (Clusiidae). Head as long as high or slightly higher than long; antennae inserted relatively distant from each from other (cf. Figure 4A and Figure 6B); arista shortly ciliate (Figure 4D and Figure 6B); only 2 dc; prescutellar ac pair short; f_2_ only slightly thicker than f_3_; R_1_ bare, without setulae; abdomen of both male and female slender, elongate; male cerci asymmetrical; gonostyli asymmetrical and bilobed. For other generic characters see definition of the family Clusiomitidae above.

**Species included**: *Clusiomites clusioides* (Roháček, 2016), *C. ornatus* sp. nov.

**Comments.** The new genus *Clusiomites* is characterized by the above diagnostic characters. However, only the following are considered putative apomorphies (with respect to states found in *Acartophthalmites*) delimiting it as a monophyletic group: antennae inserted relatively distant each from other; prescutellar ac pair short; male cerci asymmetrical; gonostyli asymmetrical and bilobed.


(1)*Clusiomites clusioides* (Roháček, 2016) comb. nov. (Figure 1A, Figure 2, Figure 3 and Figure 4)


*Acartophthalmites clusioides* Roháček, 2016: 411.

**Type material.** Holotype male, labelled: “Baltic amber, Russia: Kaliningrad region, Yantarny. Purchased from Jonas Damzen (8th ICD, Potsdam 10.-15.viii.2014)” (printed), “HOLOTYPUS ♂, Acartophthalmites clusioides sp.n., J. Roháček det. 2015” (red label), “Typ. č. 1097, Museum Silesiae” (red label, handwritten) and “Clusiomites clusioides (Roháček), J. Roháček & C. Hoffeins det. 2021” (printed), deposited in SMOC, type no. 1097; flat block-shaped amber piece ca 9.9 × 7.2 × 3.2 mm, embedded in polyester resin, size 13.8 × 10.0 × 4.9 mm (Figure 2B); syninclusions: some stellate hairs of various shapes.

**Diagnosis**. Species with bicolourous (dark brown and yellow or ochreous) body; frons surface smooth; frontal triangle short; ocelli small; oc distinctly shorter than pvt, no microsetulae on orbit and middle of frons; antenna yellowish white; scutum and thoracic pleuron with brown and ochreous pattern, but less contrasting than in *C. ornatus*; prescutellar ac setae small and at level of posterior dc; f_2_ markedly prolonged, much longer than f_3_ and ventrally with modified setosity (cf. Figure 3C); t2 with a row of seven relatively short erect posterior setae; M ending in front of wing margin; apical part of CuA1 shorter than dm-cu; A1 shorter than in all relatives (Figure 3H); epandrium short, ventrally and dorsaly subequal in length; male with right cercus more slender than left cercus; posterior part of left gonostylus long and strongly curved; anterior lobe of of right gonostylus very slender, with two setulae on apex.

**Figure 2 insects-12-01123-f002:**
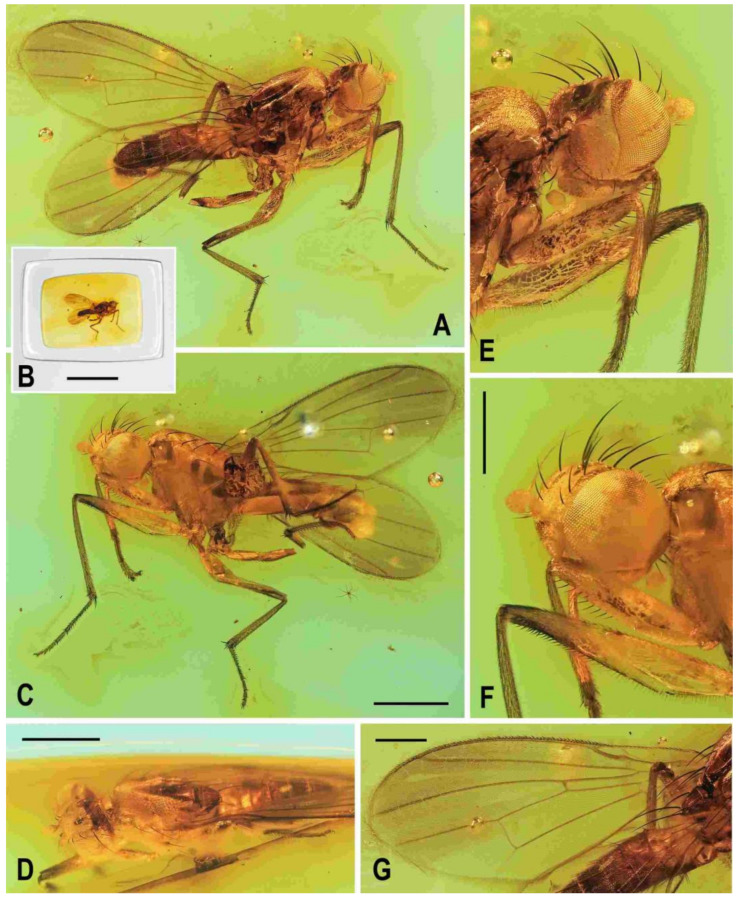
*Clusiomites clusioides* (Roháček), male holotype: (**A**) holotype, right sublaterally; (b) whole amber sample (preparatum in polyester resin), in situ; (**C**) holotype, left sublaterally; (**D**) ditto, subdorsally; (**E**) head and parts of fore and mid legs, right posterolaterally; (**F**) ditto, left anterolateraly; (**G**) left wing, dorsally. Scales: 1 mm (**A**,**C**,**D**), 5 mm (**B**) and 0.5 mm (**E**–**G**). Photos by J. Roháček.

**Description.** Male. Total body length 3.85 mm; general colour probably bicolourous, dark brown and yellow or ochreous; legs pale brown; thorax and abdominal sclerites probably subshining (Figure 1A and Figure 2A,B,D).

***Head*** (Figure 2E,F, Figure 3A and Figure 4C) about as long as high (not precisely measurable), dorsally somewhat wider than thorax. Head distinctly bicolourous, occiput darkest (dark brown), frons, face, gena and postgena pale, ochreous to whitish yellow except for some small parts. Frons relatively narrow (almost as wide as eye in dorsal view), slightly tapering anteriorly, largely yellow to ochreous with only ocellar triangle dark brown and orbits pale brown. Orbit narrowly brownish but this darkening tapered anteriorly, narrow and faded at foremost ors. Frontal triangle ochreous-brown, short (less than half length of frons) and its anterior corner identical with that of ocellar triangle. Ocellar triangle somewhat tubercle-like and elongate, protruding among small ocelli. Frontal lunule poorly visible in the specimen, probably small. Face (praefrons), parafacialia (both only partly visible) and gena yellowish white and apparently whitish microtomentose; gena with brownish line at ventral margin in addition. Postgena and adjacent part of occiput pale yellow behind eye to pale brown medially (near foramen). Cephalic chaetotaxy (Figure 3A and Figure 4C): pvt relatively strong (longer and thicker than oc); vti very long (longest cephalic seta), almost twice as long as vte; oc relatively weak (distinctly shorter than pvt) and strikingly (possibly unnaturally) erect in the holotype; three ors becoming shorter anteriorly, the hindmost ors longest and strongest (about as long as vte); no microsetulae on frons medially or in front of ors; postocular setulae dorsally in two rows surrounding posterior eye margin, none of them enlarged but there are numerous additional and relatively long erect setulae (see Figure 4C) scattered on adjacent lateral parts of occiput and postgena; postgena with two or three (one distinctly longer) posteroventral setae in addition; occiput with a brush of small setulae above foramen; vi (Figure 4D) relatively short, curved medially; peristomal setulae small and sparse (five observed). Eye strongly convex and covering most of head in profile, subcircular (slightly concave along posteroventral margin); its longest diameter oblique and only about 1.1 times as long as shortest diameter. Gena very low; its height about 0.06 times as long as shortest eye diameter. Palpus not visible, only its ventropreapical seta discernible. Mouthparts ochreous to (posterodorsally) brownish; labellum large, fleshy and finely setulose, setulae pale. Antenna (Figure 2F and Figure 3A) relatively small, yellowish white or scape + pedicel somewhat darker; pedicel externally laterally with somewhat excavated anterior margin, with one stronger erect seta dorsally and two finer setae ventrally in addition to series of marginal and submarginal setulae; first flagellomere strongly laterally compressed, in profile subcircular with excavated posterior side (Figure 4D). Arista almost 3 times as long as antenna, with elongate and whitish basal segment and darker ochreous terminal section being distinctly but shortly ciliate (Figure 4D).

**Figure 3 insects-12-01123-f003:**
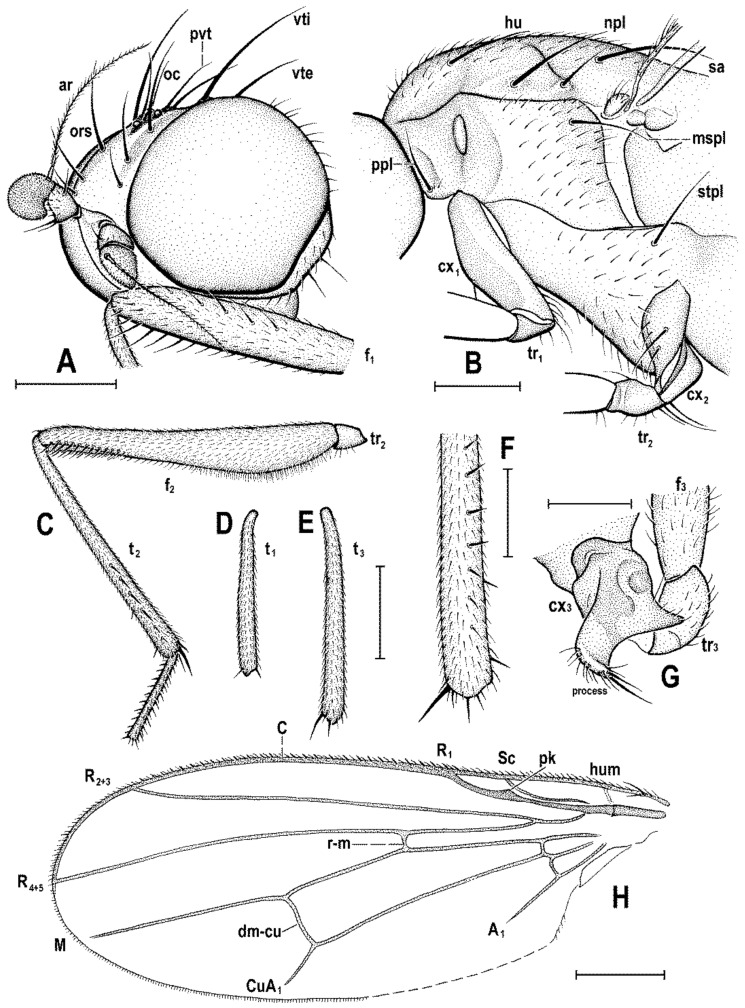
*Clusiomites clusioides* (Roháček), male holotype: (**A**) head and f_1_, left sublaterally; (**B**) thorax, left laterally; (**C**) left tr_2_, f_2_, t_2_ and mid basitarsus, posteriorly; (**D**) left t_1_, posteriorly; (**E**) left t_3_, anteriorly; (**F**) distal part of right t_2_, posteriorly; (**G**) left cx_3_, tr_3_ and base of f_3_, laterally; (**H**) left wing, dorsally. Scales: 0.3 mm (**A**,**B**), 0.5 mm (**C**–**E**,**H**) and 0.2 mm (**F**,**G**). For abbreviations see text (p. 3. Adapted from Roháček (2016: Figures 9–15, only G is original).

***Thorax*** slightly narrower than head (Figure 2D), bicolourous, largely brown, with some parts paler, ochreous (Figure 2A,C,D). Mesonotum pale brown, darker brown medially and posteriorly, with finely shagreened surface (this microstructure arranged transversely in anterior part of mesonotum). Humeral (postpronotal) callus brown and protruding; notopleural area pale brown; scutellum and subscutellum dark brown, the former with surface densely grooved longitudinally; pleural part of thorax distinctly bicolourous: anterior part of mesopleuron and almost entire sternopleuron pale ochreous; propleuron, posterior part of mesopleuron, hypopleuron (meron) and metanotum (anatergite) pale brown; pteropleuron and metapleuron dark brown. Scutellum subtriangular with rounded apex, slightly convex dorsally; subscutellum well developed. Thoracic chaetotaxy (Figure 2C,D,G, Figure 3B): one strong and long hu (plus several microsetae on humeral callus), two npl (anterior strong, twice as long as posterior; both setae broken on right side), one very long and robust sa (only slightly shorter than pa), two pa (anterior very long and strong, longer than posterior dc; posterior pa thinner and about half length of the anterior), two dc (both postsutural, also anterior situated behind level of sa), anterior short (slightly more than half length of posterior), posterior robust but shorter than apical sc or anterior pa; scutum otherwise covered by uniform and relatively dense microsetae (more than 15 dc microsetae in row in front of anterior dc); ac microsetae arranged in about eight rows in front of suture but less posteriorly, and only four rows reaching slightly beyond level of posterior dc; prescutellar ac short, only twice longer than adjacent ac microsetae and situated at level of posterior dc; two sc, apical strong and very long (with anterior pa the longest thoracic seta), also laterobasal relatively robust, as long as three-fourths of the apical sc; one long upcurved ppl plus two microsetae near its base; mesopleuron with one distinct mspl in posterodorsal corner and numerous microsetae on most of its surface (except for anteroventral part); sternopleuron with one long stpl and a number of scattered microsetae (also those on ventral part relatively short) leaving only anterodorsal corner of sclerite bare; prosternum obviously bare. 

***Legs*** originally probably all brown or pale brown (the densely brown-spotted appearance of femora is probably an artefact caused by the process of fossilization), relatively long and slender. Mid leg conspicuously prolonged while the fore and hind legs have their normal proportions. Femur, tibia and also tarsus (basitarsus in particular) of mid leg are almost twice as long as those of fore leg, and distinctly longer than those of hind leg (cf. Figure 1A and Figure 2A,C). cx_3_ with strong ventral process (Figure 3G), setose on apex; f_1_ (Figure 2F and Figure 3A) with a short row of 4–5 longer posteroventral to ventral setae in distal third and with about four posterodorsal setae forming a row in the middle third of femur; f_2_ (Figure 2F and Figure 3C) very elongate, tapered distally, finely, densely setulose but its distal fourth ventrally with a double row of thicker, shorter, denser setae (those in posterior row longer) and its proximal half with a dense brush of ventral upright hair-like setulae; f_3_ uniformly densely finely setulose. t_1_ (Figure 3D) also uniformly setulose but with a short dorsopreapical seta and a similar ventroapical one in addition; t_2_ (Figure 3C,F) besides usual short setosity with a row of about seven erect posterior setae (4–5 longer and thicker) in distal two-fifths, two short dorsal setae (1 preapical, 1 in distal third) and one longer and thicker ventroapical seta plus two shorter setae on apex; t_3_ (Figure 3E) with one longer ventroapical and one shorter anteroapical seta, otherwise uniformly finely setulose. Tarsi simple, slender; mid and hind basitarsi long and with thicker posterolaterally directed setulae. 

***Wing*** (Figure 2G and Figure 3H) distinctly elongate; veins pale brown; membrane brownish darkened in anterodistal half, having an elongate lanceolate light spot in otherwise brownish cell r_2+3_. C reaching only slightly beyond apex of R_4+5_ where attenuated. Sc fine, proximally attenuated. R_1_ bare, with distinct preapical kink (Figure 3H, pk). R_2+3_ long, very slightly sinuate, apically somewhat upcurved to C. R_4+5_ shallowly but distinctly bent posteriorly, and distally subparallel with M. Distal part of M almost straight but not reaching (ending slightly in front of) wing margin. Discal (dm) cell relatively elongate; its upper distal corner somewhat acutely projecting; anterior cross-vein (r-m) situated in about the middle of discal cell. Distal part of CuA_1_ shorter than dm-cu cross-vein and not reaching wing margin; A_1_ short, ending far from it. Cells bm and cup closed. Anal lobe moderately developed. Alula not well discernible, obviously folded (cf. Figure 3H) and probably broad although relatively small. Wing measurements: length 3.53 mm, width 1.35 mm, Cs_3_:Cs_4_ = 2.00, r-m\dm-cu:dm-cu = 2.28. Haltere pale brown, knob somewhat darker.

***Abdomen*** (Figure 2A,D and Figure 4A,B) slender, elongate, largely brown to dark brown, with dorsal sclerites elongately shagreened. All preabdominal terga rather sparsely but distinctly setose, with longest setae (some upright) at posterior and lateral margins. T1 and T2 (at least dorsally) ochreous, T3–T5 dark brown. T1 probably separate (visible only laterally) from T2. T1–T5 relatively large and long, distinctly bent laterally (pleural membrane not large). Preabdominal sterna pale brown, sparsely and short setose (only S4 and S5 visible, Figure 17) and obviously becoming wider posteriorly; S4 distinctly transverse; S5 largest and very broad, strongly transverse. Postabdomen (Figure 4A,B) with sclerites well developed, dark-pigmented and asymmetrical. Synsclerite T6 + S8 large, long and somewhat asymmetrical, particularly extended on right side (Figure 4A). S6 elongately subtriangular, ventrally attenuated, situated on left side of postabdomen (Figure 4B) and probably (not precisely visible) attached to or fused with T6 + S8 dorsally and, at least partly, also to S7; S7 probably also subtriangular but its shape is not precisely discernible in the holotype. S6 apparently bare but S7 with at least one setula (most of sclerite covered by white coating so that its setosity cannot be recognized). T6 + S8 rather densely short setose but setae are only visible on its right side (Figure 4A).

**Figure 4 insects-12-01123-f004:**
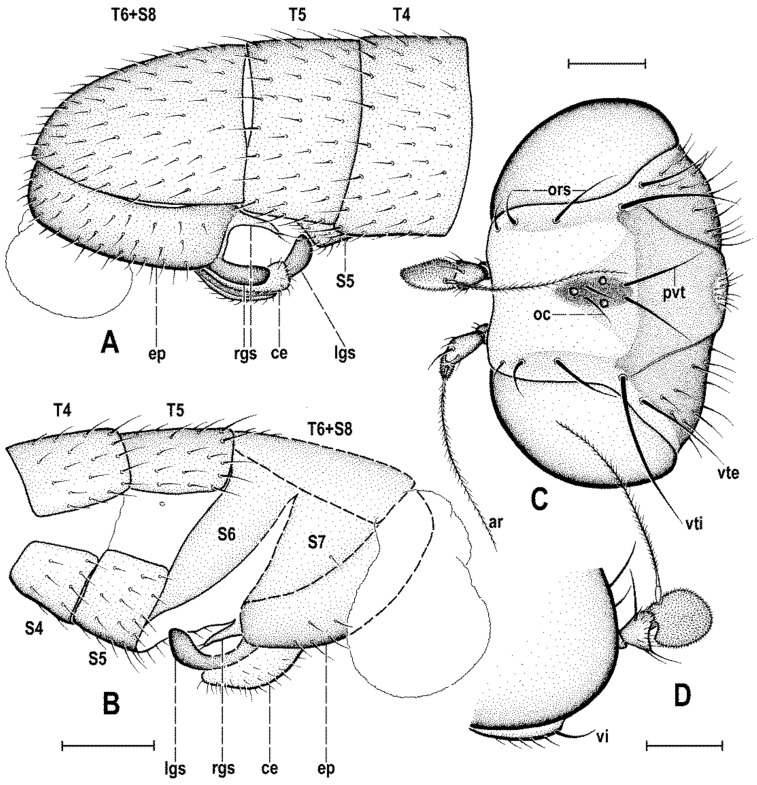
*Clusiomites clusioides* (Roháček), male holotype: (**A**) end of abdomen with external genitalia, right laterally; (**B**) ditto, left laterally; (**C**) head, subdorsally; (**D**) anteroventral part of head with antenna, right laterally. Scales: 0.2 mm. For abbreviations see text (p. 3. Adapted from Roháček (2016: Figures 16–19).

***Genitalia***. Epandrium (Figure 4A,B) short and high, with anal fissure directed ventrally, probably entirely shortly uniformly setose (only a few setae visible on left side). Anal fissure cannot be seen in the holotype. Gonostyli (Figure 4A,B) relatively small but clearly asymmetrical. Right gonostylus (Figure 4A) strongly bilobed, with anterior lobe very narrow and pale-pigmented, distinctly pointed and with a pair of small setae on apex; posterior lobe digitiform, dark, heavily sclerotized and bare, apically rounded. Left gonostylus (Figure 4B) with posterior lobe curved and distally dilated, markedly longer than that of right gonostylus, but its anterior lobe not observable or absent. Cerci also asymmetrical (cf. Figure 4A,B) and situated ventrally between posterior lobes of gonostyli and projecting below ventral epandrial margin; left cercus with apex in profile more rectangular, right cercus larger and with apex more rounded, both pale-pigmented and finely haired. Internal genitalia (aedeagal and/or hypandrial complex) not visible.

Female. Unknown.

**Comments**. *Clusiomites clusiosides* Roháček, 2016 is an easily recognizable species, differing from all other Clusiomitidae by its slender body and very elongated f_2_ with modified ventral setosity. Its wing venation is also somewhat modified, including uniquely shortened M and A_1_, not to mention the characteristic male terminalia.


(2)*Clusiomites ornatus* sp. nov. (Figure 5, Figure 6, Figure 7 and Figure 8)


LSID urn:lsid:zoobank.org:act:C094E00D-82FC-4763-958A-AEDE7D

**Type material.** Holotype male, labelled: “Faszination Bernstein, Christel Hoffeins, Hans Werner Hoffeins” (framed on obverse), ”1710-1 Acartophthalmidae, Acartophthalmites ♂” (handwritten by C. Hoffeins, on reverse), ”Russia: Kaliningrad Region, Baltic Sea coast, Yantarny” (printed) and “Holotypus ♂, Clusiomites ornatus sp.n., J. Roháček & C. Hoffeins det. 2021” (red label), deposited in CCHH, no. 1710-1 [block-shaped amber piece ca 10 × 8 × 4 mm, embedded in polyester resin, size 12.9 × 11.5 × 6.7 mm] (Figure 5G); syninclusions: minute plant remnants only.

Paratype female, labelled “Faszination Bernstein, Christel Hoffeins, Hans Werner Hoffeins” (framed on obverse), “897-7 Acartophthalmidae, Acartophtalmites tertiaria ♀” (handwritten by C. Hoffeins, on reverse), “Russia: Kaliningrad Region, Baltic Sea coast, Yantarny” (printed) and “Paratypus ♀, Clusiomites ornatus sp.n., J. Roháček & C. Hoffeins det. 2021” (yellow label), deposited in CCHH, no. 897-1 [flat irregular block-shaped amber piece ca 14 × 11 × 3 mm, embedded in polyester resin, size 16.9 × 15.8 × 5.2 mm] (Figure 7E); syninclusions: remnant of insect wing and leg (Lepidoptera), stellate hairs, dirt.

**Type locality:** Russia: Kaliningrad Region, Baltic Sea coast, Yantarny (formerly Palmnicken). Note: all amber samples with type specimens were purchased from commercial sources in Lithuania and Poland who got the material from Yantarny.

**Etymology:** The name of the new species “ornatus” (=adorned, ornamented, from Latin, adjective) refers to bicolourous ornamentation of its thorax.

**Diagnosis.** Species with bicolourous (dark brown and ochreous to pale yellow) body and legs; frons surface smooth; frontal triangle relatively long; ocelli large; oc as long as pvt; orbit and also middle of frons with additional microsetulae; palpus brown; antenna pale brown to ochreous-yellow; scutum and thoracic pleuron with contrasting pattern (pleuron dark brown, only sternopleuron pale yellow ventrally); prescutellar ac setae medium-long and situated beyond posterior dc; f_2_ only slightly thicker and about as long as f_3_ and ventrally simply setulose; t_2_ with a row of eight or nine relatively short erect posterior setae; M reaching wing margin; apical part of CuA_1_ slightly longer (1.1 times as long as) than dm-cu; A_1_ long; epandrium relatively long, ventrally longer than dorsally. Male cerci strongly asymmetrical, right cercus broad and shorter, left cercus slender and long; posterior part of left gonostylus long and slender but less curved than that of *C. clusioides*; anterior lobe of right gonostylus broad, subtriangular; female S8 bipartite.

**Figure 5 insects-12-01123-f005:**
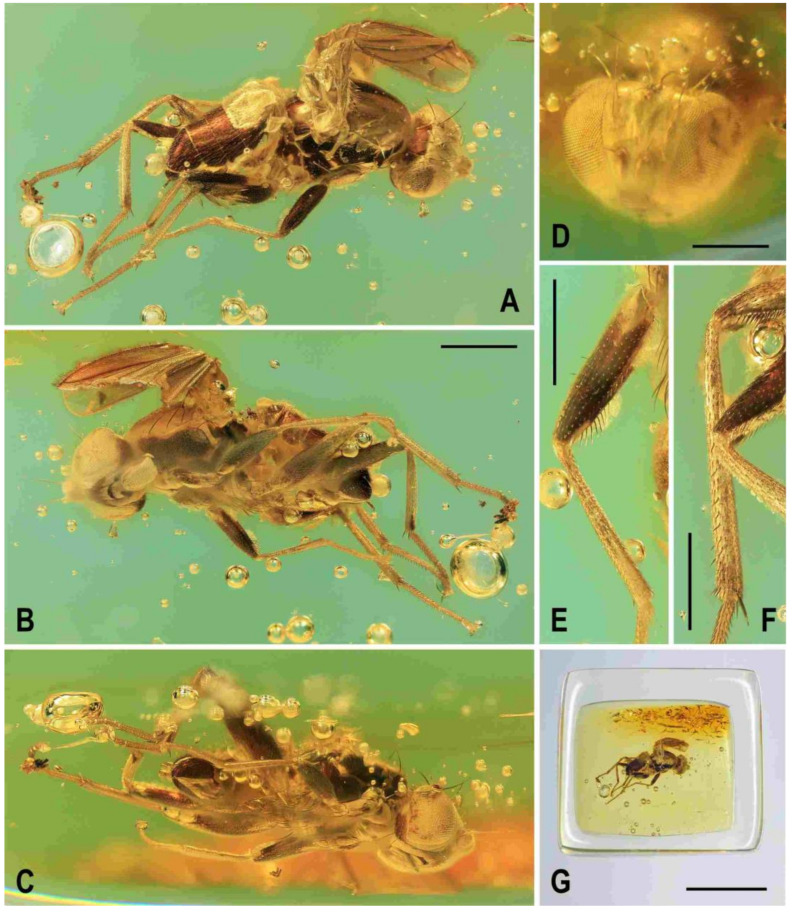
*Clusiomites ornatus* sp. nov., male holotype: (**A**) holotype, right dorsolaterally; (**B**) ditto, left ventrolaterally; (**C**) ditto, subventrally; (**D**) head, frontally; (**E**) right f_1_ and t_1_, posteriorly; (**F**) left t_2_ posteriorly; (**G**) whole amber sample (preparatum in polyester resin), in situ. Scales: 1 mm (**A**–**C**), 0.5 mm (**D**–**E**), 5 mm (**G**). Photos by J. Roháček.

**Description.** Male. Total body length about 4.0 mm; body bicolourous, dark brown and pale ochreous-yellow variegated; legs brown and yellow, bicolourous.

***Head*** (Figure 5A,C,D and Figure 6B) about as long as or slightly longer than high; occiput dorsally slightly concave. Head bicolourous, brown or pale brown and whitish yellow. Occiput largely brown but pale ochreous margined along eyes. Frons pale brown and whitish yellow, relatively narrow (narrower than in *C. clusioides)* but wider than eye in dorsal view, slightly tapering anteriorly, smooth on surface. Orbit with pale brown (darker posteriorly) stripe being anteriorly tapered, otherwise whitish yellow along eye. Medial part of frons (between frontal triangle and orbits) light ochreous to whitish yellow (anteriorly). Frontal triangle pale brown to pale ochreous, dull, relatively long, with anterior corner acute and reaching to anterior fourth or fifth of frons; ocellar triangle darker (brown) and somewhat protruding; ocelli relatively large and closely situated. Frontal lunule poorly visible, probably short and somewhat depressed, whitish yellow as is adjacent part of frons. Face largely brown, narrow, flat to very slightly convex; parafacialia and gena dirty white but brown to dark brown narrowly margined. Postgena and adjacent part of occiput relatively large, both brown but with ochreous yellow stripe along eye. Cephalic chaetotaxy (Figure 6B) with macrosetae relatively long but thin: pvt about as long as vte; vti strong and long (longest cephalic seta), about 1.5 times as long as vte; oc relatively long (as long as pvt) but thinner, divergent and proclinate; three relatively short and fine ors becoming distinctly shorter anteriorly, hindmost ors longest but shorter than vte or pvt; a few additional microsetulae on orbit and one or two also medially near anterior corner of frontal triangle; postocular setulae rich and relatively long, in two or three rows dorsally, plus more additional setulae on adjacent parts of occiput and postgena; postgena with three fine but rather long posteroventral setae in addition; vi short but distinct, thicker and more than twice as long as foremost peristomal setula (vibrissae curved medially, with apices crossed); peristomal setulae numerous (9–10 visible). Eye large and relatively long, broadly subovoid (with posterior margin slightly excavated), its longest diameter less than 1.2 times as long as shortest diameter. Gena very low, its height only 0.07 times as long as shortest eye diameter. Palpus brown, small but relatively long (Figure 5B,C), with distinct black setulae (one longer subapical, four or five distinct dorsal and several shorter lateral and ventral). Mouthparts ochreous but clypeus blackish brown; labellum pale ochreous, fleshy and finely pale haired. Antenna pale brown to ochreous-yellow (scape and pedicel darker) and relatively small; pedicel externolaterally with anterior margin somewhat convex dorsally, with one longer seta dorsally and one finer seta ventrally apart from marginal and submarginal setulae; first flagellomere distinctly longer than pedicel, laterally compressed and suboval (higher than long) in profile, entirely shortly finely whitish pilose. Arista about 2.7 times as long as antenna, with shortly cylindrical ochreous basal segment and somewhat darker terminal section having short and fine ciliation (Figure 6B).

***Thorax*** (Figure 5A–C). Somewhat narrower than head, with bicoulorous pattern. Scutum largely pale ochreous but with brown to blackish brown medial band being anteriorly attenuated and posteriorly almost as broad as scutellum; scutellum and subscutellum blackish brown; pleural part of thorax largely blackish brown but sternopleuron ventrally yellow to whitish yellow; dark pleural sclerites more or less distinctly pale margined; sternal part of thorax (including prosternum) whitish yellow. Scutellum relatively long (1.5 times as broad as long), subtriangular and distinctly convex dorsally; subscutellum slightly bulging. Thoracic chaetotaxy (cf. also Figure 7A,B): all macrosetae relatively thin; one hu (almost as long anterior npl) and several microsetae on humeral callus; two npl, anterior distinctly longer than posterior; 1 strong sa, longer than anterior npl; two pa, anterior very long and strong (longest thoracic seta), posterior fine and one-third to one-half of anterior; two dc, anterior dc situated at level of sa and about half length of posterior; posterior dc long but only slightly longer than laterobasal sc; one prescutellar medial ac distinct, situated beyond level of posterior dc, finer and shorter than posterior pa; scutum covered by microsetae becoming distinctly smaller and denser anteriorly; ac microsetae in about 10–12 irregular rows in front of suture but in less rows posteriorly (six rows reaching level of posterior dc); two sc, apical, very long but somewhat shorter than anterior pa, laterobasal sc thinner and shorter (about 0.7 times as long as) apical sc; one long and relatively strong (subequal to hu) upcurved ppl; mesopleuron with one distinct mspl (somewhat shorter than hu) in posterodorsal corner and numerous scattered microsetae on most of sclerite (those at posterior margin longer); sternopleuron with one long (longer than mspl) stpl and a several microsetae scattered over all surface, and leavingtwo or three longer setae also on ventral corner of sternopleuron.

**Figure 6 insects-12-01123-f006:**
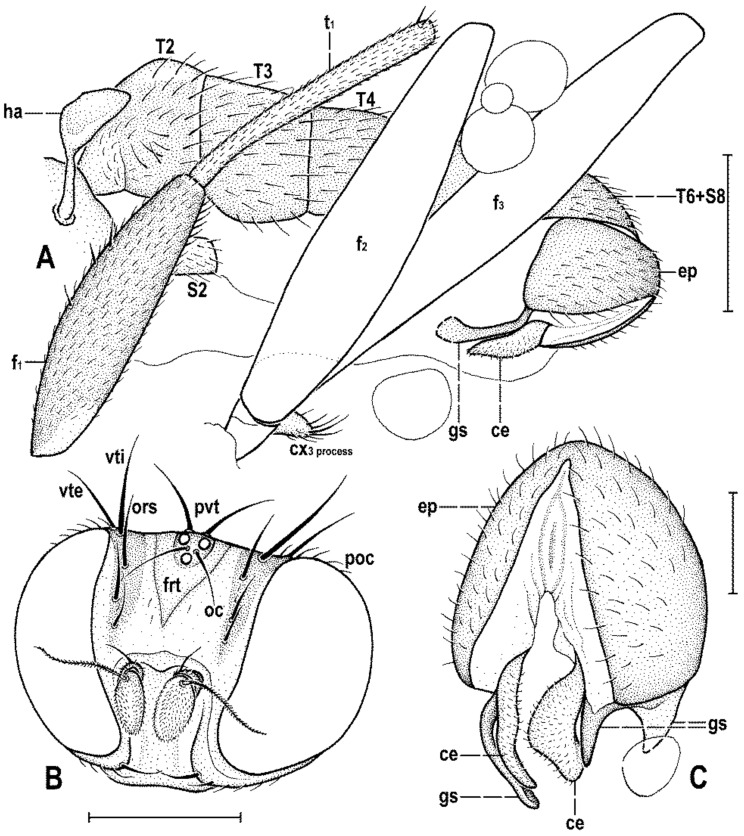
*Clusiomites ornatus* sp. nov., male holotype: (**A**) abdomen and parts of left legs, left laterally; (**B**) head, frontally; (**C**) external genitalia, caudally. Scales: 0.5 mm (**A**,**B**) and 0.2 mm (**C**). For abbreviations see text (p. 3).

***Legs*** (Figure 5A,B,E,F) bicolourous: coxae ochreous to brownish spotted; trochanters whitish yellow; femora dark brown but their bases (that of f_1_ only ventrally, see Figure 6A) pale yellow and knee of f_1_ ochreous; tibiae and tarsi pale yellow but apices of t_2_ and, particularly t_3_ more or less brownish darkened. Femur and tibia of mid leg about 1.7 times as long as those of fore leg; f_1_ relatively thick although less than f_2_ or f_3_; both f_2_ and f_3_ (see Figure 5A,B) robust, thickened and subequal in length; cx_3_ distally with ventral process having a tuft of setae on apex; f_1_ with short but dense row of 7–8 posteroventral setae in distal fifth (Figure 5E) and with two or three short erect posterodorsal setae in row in middle fourth of femur; f_2_ elongated and thickened but in proximal third only slightly thicker than f_3_ in this place; both f_2_ and f_3_ without specific setae, uniformly finely setulose; t_1_ (Figure 6A) finely setulose but with 1 pair of small dorsal preapical setae; t_2_ with a row of eight or nine erect but relatively short posterior setae (those in the middle longest, cf. Figure 5F and Figure 8D) in distal three-fifths, three small dorsal setae (one preapical, others in distal fifth) and one long and robust ventroapical seta plus a series of smaller but thick setae (1 anterior longer, half length of ventroapical seta) on apex around it; t_3_ straight (Figure 5B), with one short ventroapical seta (distinctly shorter than that on t_2_), one subequal anteroapical seta and a few thicker setulae on apex. Tarsi simple, slender; fore basitarsus with two longer fine setulae ventrobasally; mid and hind basitarsi long (longer than rest of tarsus) and each with a row of thicker posteroventral erect setulae.

***Wing*** largely undescribed because severely damaged (twisted) in holotype (cf. Figure 5A,B). For wing description see below under female. Wing distinctly elongate; veins ochreous to pale brown, membrane distinctly brownish fumose, particularly so at anterior margin and on apex. C attenuated beyond R_4+5_, but this slender nebulous part reaching to M; C finely setulose but Cs_2_ and partly also Cs_3_ (only very basally) with thicker, short spine-like setulae in addition. R_1_ short, wholly bare, with distinct preapical kink at the level of apex of Sc. Alula large and broad (Figure 5B). Upper calypter rounded and with a tuft of long hair-like setae. Wing measurements: not measurable except for Cs_3_:Cs_4_ = 1.43. Haltere dirty white, at most stem somewhat darker.

***Abdomen*** (Figure 5A–C and Figure 6A). Bicolourous, subcylindrical, elongately pyriform in dorsal view (broadest at 3rd segment), very slightly bent in lateral view (Figure 6A). All preabdominal terga relatively densely and short setose, with longest setae at posterior margins; setae on T1 proximally small and somewhat erected, also some (longer) lateral setae in the middle of T2 upright. T1 entirely, T2 and T3 only dorsally pale ochreous yellow, the latter two laterally brown to dark brown. T4 and T5 entirely dark brown. T2–T5 very narrowly pale margined posteriorly. T1 only laterally separate from T2. T1 shortest and narrow, T2–T5 subequal in length. Preabdominal sterna only partly visible. S2 apparently narrow and pale yellow, shortly setulose. S4 and S5 pale brown, subequal in length, slightly transverse and shortly setulose. Postabdomen with only large synsclerite T6 + S8 visible (see Figure 5A and Figure 6A), the latter almost twice longer than T5, tapered posteriorly and somewhat asymmetrical, larger on left side, blackish brown (darker than T5) but with narrowly pale posterior margin, densely short setose. 

***Genitalia***. Epandrium blackish brown, short and high (ventrally longer, dorsally tapered), with anal fissure directed ventrally (Figure 5C and Figure 6A), shortly uniformly setulose (setosity distinctly shorter and finer that that of T6 + S8) except for anterior third. Annal fissure (Figure 6C) long (=high), elongately triangular, with dorsal corner very acute; epandrium ventrally open. Cerci distinctly separate (Figure 6C), strongly asymmetrical (left slender and longer, right broad and shorter), both finely setulose, each inserted at posteroventral margin of epandrium close to posterior part of gonostylus. Gonostyli (Figure 6A,C) darker than cerci, almost bare, both obviously bilobed and also distinctly asymmetrical: posterior part of left gonostylus long, slender and distally club-shaped; right gonostylus with posterior part short and distally simply tapered; its anterior part paler, and probably acutely subtriangular (apex not visible).

Female. Similar to male unless mentioned otherwise. Larger, total body length ca 5.0 mm. ***Head***. Chaetotaxy as in male but all setae more robust. Postocular setulae somewhat longer and more numerous; up to three pairs of additional microsetulae medially near apex of frontal triangle. Palpus dark brown and with more black setulae. Clypeus brown. Antenna darker (brown including 1st flagellomere). ***Thorax***. Scutum, apart from medial dark brown band, with smaller and short dark lateral stripe between dc and sa-pa lines (Figure 7A,B). Sclerites of pleuron with wider pale margins; sternopleuron more extensively yellowish white ventrally (Figure 7B,C). Prescutellar ac pair longer and situated slightly beyond posterior dc. Legs. Coxae paler. f_1_ (Figure 7B and Figure 8C) basally more extensively whitish (as are f_2_ and f_3_) and with four or five short erect posterodorsal setae.

**Figure 7 insects-12-01123-f007:**
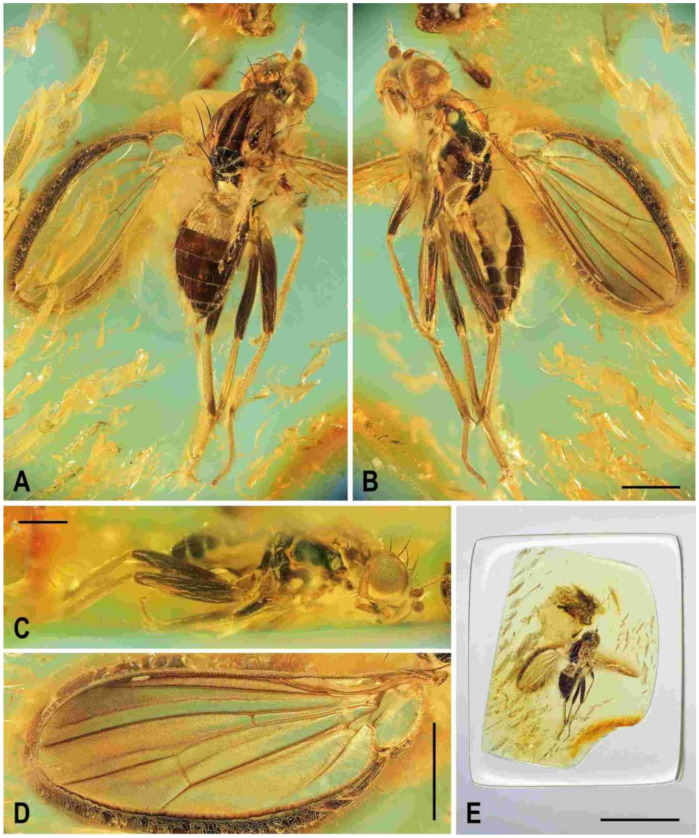
*Clusiomites ornatus* sp. nov., female paratype: (**A**) paratype, right subdorsally; (**B**) ditto, left subventrally; (**C**) ditto, right laterally; (**D**) left wing, dorsally; (**E**) whole amber sample (preparatum in polyester resin), in situ. Scales: 1 mm (**A**–**D**) and 5 mm (**E**). Photos by J. Roháček.

***Wing***. Relatively long and narrow (Figure 7D); membrane brownish darkened along anterior margin and in apical third. C finely setulose but Cs_2_ with small and sparse spine-like setulae, last of which are situated very slightly beyond apex of R_2+3_. Sc free and well developed. R_1_ robust, lacking setulae but with distinct (and darkened) preapical kink. R_2+3_ long, very slightly sinuate, running close to wing margin, apically slightly upcurved to C and ending far from wing apex. R_4+5_ slightly but distinctly bent posteriorly, distally straightened and ending close to wing apex. Distal part of M almost straight, parallel with R_4+5_ and reaching wing margin. Discal (dm) cell large and long; anterior cross-vein (r-m) situated near middle of cell. Distal part of CuA_1_ relatively short, about 1.1 times as long as dm-cu and almost reaching wing margin; A_1_ long, but ending far from wing margin. Alula large and very broad (Figure 7D). Wing measurements: length 4.29 mm, width 1.59 mm, Cs_3_:Cs_4_ = 1.96, r-m\dm-cu:dm-cu = 2.50.

**Figure 8 insects-12-01123-f008:**
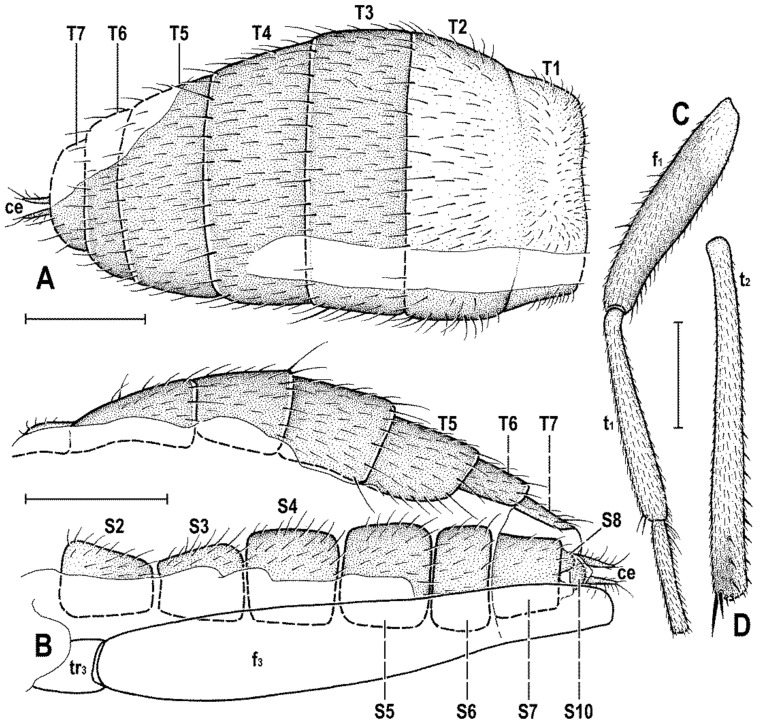
*Clusiomites ornatus* sp. nov., female paratype: (**A**) abdomen dorsally; (**B**) ditto, subventrally (partly reconstructed); (**C**) right f_1_, t_1_ and fore basitarsus, posteriorly; (**D**) left t_2_, anterodorsally. Scales: 0.5 mm. For abbreviations see text (p. 3).

***Abdomen*** (Figure 7A–C and Figure 8A,B) bicolourous basally as in male, relatively narrow, elongate, widest at third segment, then gradually tapered posteriorly (Figure 8A). Preabdominal terga (Figure 7A and Figure 8A) somewhat shorter and more transverse than in male, all densely setose, with longest setae at posterior margins. In contrast to male, only T1 and T2 dorsally pale ochreous and both laterally dark brown; T3–T5 entirely dark brown. T1 only laterally separate from T2; T2 somewhat longer than T3; T3–T5 subequal in length; T2 widened posteriorly; T3 widest tergum; T4 and T5 becoming narrower posteriorly, hence T5 distinctly narrower and less transverse than T4. Ventral side of abdomen partly visible (see Figure 7B and Figure 8B). Preabdominal sterna (S2–S5) shortly and finely setose, all relatively narrow. S1 not discernible; S2 wider anteriorly and narrower posteriorly; S3 as broad as S2 anteriorly and both about as long as broad; S2 paler brown than S3, S3–S5 brown. S4 distinctly wider than S3 but narrower than S5, both subequal in length and very slightly transverse. S5 widest preabdominal sternum.

***Postabdomen*** (Figure 8A,B). T6 only less than half length of T5, somewhat narrower and distinctly transverse; T7 slightly longer but markedly narrower than T6, both dark brown and sparsely setose. T8 and T10 not visible. S6 shorter and slightly wider than S5, transverse; S7 distinctly longer than S6, slightly tapered posteriorly; both S6 and S7 brown to dark brown. S8 small and short, apparently medially divided in two parts being finely sparsely setulose; S10 small, semicircular, finely shortly setulose (Figure 8B). Cerci pale brown, slender and setose as in relatives.

**Comments**. *Clusiomites ornatus* is a markedly bicoloured species, having dark brown femora and the rest of legs yellowish. It can also be separated from its congener, *C. clusioides* Roháček, 2016, by the unusually long oc (as long as pvt), relatively large frontal triangle, long M and A_1_, and differently formed male cerci and gonostyli. Because the postabdominal sterna are visible in the female paratype, we can state that they are (probably in all Clusiomitidae) markedly broader than in the related Clusiidae (where they are very elongate and slender), but as in Clusiidae, S8 seems to be medially divided.

#### 3.1.3. Genus ***Acartophthalmites*** Hennig, 1965

*Acartophthalmites* Hennig, 1965: 130; Roháček, 2016: 419 (supplement of description).

Type species: *Acartophthalmites tertiaria* Hennig, 1965 (original designation).

**Diagnosis.** Body more robust than in *Clusiomites* species, particularly in female (Figure 9A,B and Figure 24A,B). Head distinctly shorter than high; antennae inserted close to one another (cf. Figures 11A and 16B); arista with longer ciliation (Figures 16E, 20A and 26D); two or three dc; prescutellar ac pair relatively long; f_2_ distinctly thicker than f_3_; R_1_ setulose; abdomen of female broad, subovoid; at least male cerci symmetrical; gonostyli probably simple and more or less symmetrical. For other generic characters see definition of the family Clusiomitidae above.

**Species included**: *Acartophthalmites crassipes* sp. nov., *A. luridus* sp. nov., *A. rugosus* sp. nov., *A. tertiaria* Hennig, 1965, *A. willii* Pérez-de la Fuente, Hoffeins et Roháček, 2018

**Comments.** The genus *Acartophthalmites* differs from *Clusiomites* by all above characters but only the following are treated as putative apomorphies supporting its monophyly: arista with longer ciliation; R_1_ setulose; f_2_ distinctly thicker than f_3_; abdomen of female broad, subovoid.

**Preimaginal stages** (based on female *Acartophthalmites* sp. cf. *crassipes,*
Figure 9A,B).

**Material examined.** Female, labelled: “Faszination Bernstein, Christel Hoffeins, Hans Werner Hoffeins” (framed on obverse), ”1663-3 Acartophthalmidae, A. tertiaria ♀ + egg” (handwritten by C. Hoffeins, on reverse), ”Russia: Kaliningrad Region, Baltic Sea coast, Yantarny” (printed) and “Acartophthalmites sp. cf. crassipes, ♀ + egg, J. Roháček & C. Hoffeins det. 2021”, deposited in CCHH, no. 1663-3 (block-shaped amber piece ca 8.2 × 8 × 4.2 mm, embedded in polyester resin, size 15 × 10.7 × 7.1 mm); syninclusions: 1 stellate hair, ball-shaped grains (insect faeces?).

***Egg*** (Figure 9B–D). Length 0.7 mm, maximum width 0.27 mm. Elongate subovoid, widest at posterior third, with anterior end more tapered, posterior end wider and more rounded. Dorsal surface more convex than ventral. Surface of chorion (Figure 9D) yellowish white, somewhat translucent, with rather coarse sculpture formed by irregular longitudinal rows of small very densely arranged and deep pits. Micropyle obviously situated at anterior (more tapered) end.

**Figure 9 insects-12-01123-f009:**
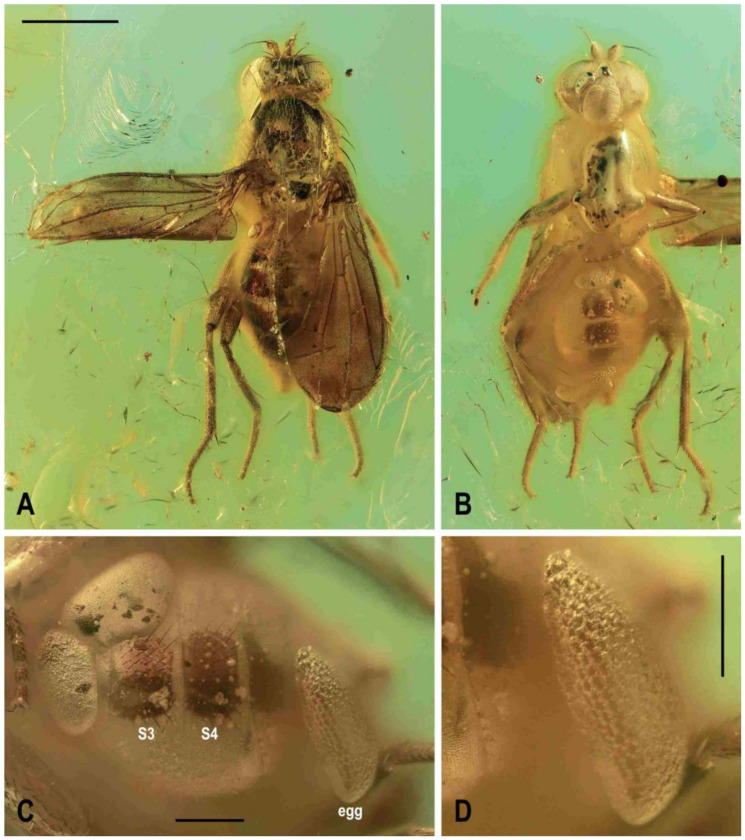
*Acartophthalmites* sp. cf. *crassipes*, female: (**A**) whole specimen, dorsally; (**B**) ditto, with egg near end of abdomen, ventrally; (**C**) abdomen with egg, ventrally; (**D**) egg (enlarged), laterally. Scales: 1 mm (**A**,**B**), 0.3 mm (**C**,**D**). Photos by J. Roháček. For abbreviations see text (p. 3).


(1)*Acartophthalmites crassipes* sp. nov. (Figure 1B, Figure 10, Figure 11, Figure 12 and Figure 13)


LSID urn:lsid:zoobank.org:act:1F4E9C4F-5379-4D87-800B-A9A29D8E2A4B

**Type material.** Holotype male, labelled: “Faszination Bernstein, Christel Hoffeins, Hans Werner Hoffeins” (framed on obverse), ”1663-5 Acartophthalmidae, A. tertiaria ♂” (handwritten by C. Hoffeins, on reverse), ”Russia: Kaliningrad Region, Baltic Sea coast, Yantarny” (printed) and “Holotypus ♂, Acartophthalmites crassipes sp.n., J. Roháček & C. Hoffeins det. 2021” (red label), deposited in CCHH, no. 1663-5 (block-shaped amber piece ca 6.7 × 6 × 3 mm, embedded in polyester resin, size 11.7 × 10.3 × 5.1 mm) (Figure 10G); syninclusions: pollen grains.

Paratype female, labelled: “Faszination Bernstein, Christel Hoffeins, Hans Werner Hoffeins” (framed on obverse), ”1175-1b Acartophthalmidae, A. tertiaria” (handwritten by C. Hoffeins, on reverse), ”Russia: Kaliningrad Region, Baltic Sea coast, Yantarny” (printed) and “Paratypus ♀, Acartophthalmites crassipes sp.n., J. Roháček & C. Hoffeins det. 2021” (yellow), deposited in CCHH, no. 1175-1b (block-shaped amber piece ca 14 × 9 × 6 mm, embedded in polyester resin, size 17.8 × 11.9 × 7.4 mm) (Figure 12C); syninclusions: Diptera: Mycetophilidae, 1 female; Coleoptera: Latridiidae, 1 specimen; insect faeces. The amber with paratype was originally a part of taphocoenosis that was cut into three pieces. Syninclusions in #1175-1a: fragment of Thuja like conifer, Diptera: Dolichopodidae, male and Hybotidae, female; Collembola, two Acari phoretic on Dolichopodidae; Myriapoda: Synxenidae; deposited at GZG. Syninclusions in #1175-1c: Diptera: Keroplatidae, female and two Cecidomyiidae; Coleoptera: ?Latridiidae, two specimens; Hymenoptera: Scelionidae, female; Homoptera: Coccoidea crawler; stellate hairs.

Paratype female, labelled “Faszination Bernstein, Christel Hoffeins, Hans Werner Hoffeins” (framed on obverse), ”897-3 Acartophthalmidae, A. tertiaria ?♂ (handwritten by C. Hoffeins, on reverse), ”Russia: Kaliningrad Region, Baltic Sea coast, Yantarny” (printed) and “Paratypus ♀, Acartophthalmites crassipes sp.n., J. Roháček & C. Hoffeins det. 2021” (yellow label), deposited in CCHH, no. 897-3 (block-shaped amber piece ca 10 × 6.5 × 4.6 mm, embedded in polyester resin, size 15 × 8.8 × 5.4 mm) (Figure 13E); syninclusions: 1 part of insect leg (?tibia), several tufts of twisted plant hairs.

**Type locality:** Russia: Kaliningrad Region, Baltic Sea coast, Yantarny (formerly Palmnicken). Note: all amber samples with type specimens were purchased from commercial sources in Lithuania and Poland who got the material from Yantarny.

**Etymology:** The species is named “crassipes” (=thick leg, from Latin, noun in apposition) owing to strongly thickened mid femur of both sexes.

**Diagnosis.** Dark blackish brown species with uniformly dark brown wings; frons surface smooth; frontal triangle small and short; orbit with additional microsetulae; first antennal flagellomere distinctly paler than pedicel; scutum without contrasting pattern; two dc; prescutellar ac setae distinctly beyond posterior dc; laterobasal sc distinctly shorter (0.7 times as long as) than apical sc; f_2_ strongly swollen and slightly longer than f_3_; t_2_ with a long row of 10–12 erect posterior setae; apical part of CuA_1_ 1.3 times as long as dm-cu.

**Description.** Male. Total body length about 3.1 mm (holotype); general colour blackish brown to brown, including thoracic pleuron (Figure 1B and Figure 10A–C).

***Head*** (Figure 10E,F and Figure 11A). Brown anteriorly to blackish brown posteriorly, occiput darkest; frons dark brown to brown; face and gena blackish brown, postgena brown. Frons relatively narrow, about as broad as eye in dorsal view, slightly tapering anteriorly, smooth on surface. Orbit more or less distinctly delimited, darker posteriorly, lighter brown anteriorly. Frontal triangle small and narrow, with anterior corner acute and not reaching half of frons; ocellar triangle black, half length of frontal triangle, distinctly protruding; ocelli of moderate size. Frontal lunule small, somewhat depressed, blackish brown. Face narrow, parafacialia distinct. Postgena and adjacent part of occiput relatively large. Cephalic chaetotaxy (Figure 11A): pvt strong (as long as vte); vti strong and long (longest cephalic seta), about twice as long as vte; oc relatively robust (as long as foremost ors); three ors becoming shorter anteriorly, hindmost ors longest and strongest (distinctly longer than vte but shorter than vti); 3–4 additional microsetulae on orbit, one each between ors, 1–2 in front of anterior ors; postocular setulae in one row dorsally at posterior eye margin but in two rows in ventral two-thirds of occiput plus some erect additional setulae scattered on adjacent medial part of occiput and postgena; postgena with two or three posteroventral setae (hindmost longest) in addition; vi relatively short but distinct, thicker and about 2.5 times as long as foremost peristomal setulae (vibrissae curved, with apices meeting medially); peristomal setulae small and fine (five or six visible). Eye broadly subovoid (with posterior margin slightly excavated), its longest diameter about 1.5 times as long as shortest diameter. Gena very low, its height only 0.06 times as long as shortest eye diameter. Palpus brown, small and short, with one black subapical setula and pale microsetae on apex. Mouthparts ochreous; clypeus pale brown; labellum lightest and fleshy, finely, pale and relatively long-haired. Antennae closely inserted, anteroventrally directed (see Figure 10F) and relatively small; scape and pedicel brown; pedicel externo-laterally with anterior margin slightly convex, with one longer seta dorsally and two or three finer setae ventrally besides marginal and submarginal setulae; first flagellomere contrastingly ochreous-orange, longer than pedicel, laterally compressed and suboval (slightly higher than long) in profile, finely whitish pilose. Arista relatively short, 2–2.1 times as long as antenna, with shortly cylindrical ochreous basal segment and dark brown terminal section being moderately long-ciliate (Figure 10F and Figure 11A).

**Figure 10 insects-12-01123-f010:**
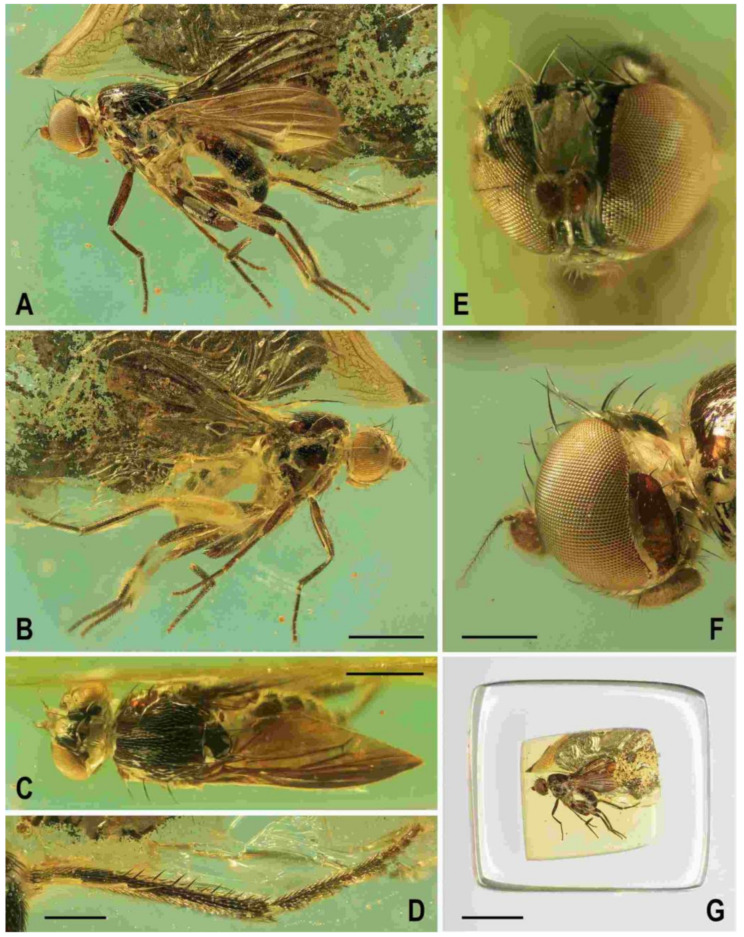
*Acartophthalmites crassipes* sp. nov., male holotype: (**A**) holotype, left laterally; (**B**) ditto, right laterally; (**C**) ditto, dorsally; (**D**) right t_2_, subdorsally; (**E**) head, frontally; (**F**)–ditto, left laterally; (**G**) whole amber sample (preparatum in polyester resin), in situ; Scales: 1 mm (**A**–**C**), 0.3 mm (**D**–**F**), 3 mm (**G**). Photos by J. Roháček.

***Thorax*** (Figure 1B and Figure 10A–C). Very slightly narrower than head, largely blackish brown. Scutum blackish brown, not patterned, at most somewhat paler laterally and posteriorly, with notopleural area brown; scutellum very dark, almost black; subscutellum probably brown; pleural part of thorax dorsally darkest (distinctly darker than adjacent notopleural area) and gradually becoming lighter ventrally. Scutellum relatively large, subtriangular with rounded apex, slightly convex dorsally; subscutellum poorly visible. Thoracic chaetotaxy (Figure 10A,C): 1 hu (strong and as long as anterior npl); two npl, anterior slightly longer but thicker than posterior; one robust sa, longer than anterior npl; two pa, anterior very long and strong (longest thoracic seta), posterior fine and less than one-third of anterior; two dc (both postsutural), anterior situated at level of sa and about two-thirds of length of posterior, posterior dc robust but only as long as laterobasal sc; one microseta in front of anterior dc slightly enlarged; one prescutellar medial ac (situated distinctly beyond level of posterior dc) subequal in length and thickness to posterior pa; scutum otherwise covered by relatively dense microsetae becoming distinctly smaller anteriorly; ac microsetae in about 10 rows in front of suture but less posteriorly (only four rows reaching level of posterior dc); two sc, both strong, apical very long but slightly shorter than anterior pa, laterobasal sc slightly thinner and shorter (0.7 times as long as) apical sc; one long but fine upcurved ppl; mesopleuron with one distinct mspl (as long as hu but thinner) in posterodorsal corner and scattered (not very numerous) microsetae in posterodorsal half of the sclerite, some of those at posterior margin being longer; sternopleuron with one long (longer than mspl) stpl and a several microsetae scattered in posterior two-thirds of surface, one or two longer but fine setae also on ventral corner of sternpleuron.

***Legs*** (Figure 10A,B,D). Brown to dark brown including tarsi, mid and hind legs long and robust. Femur and tibia of mid leg about 1.7–1.8 times as long as those of fore leg; f_2_ (see Figure 10A) strongly swollen and slightly longer than f_3_; cx_3_ distally projecting and finely setose but this process probably not acute (not precisely seen); f_1_ with a short row of 5–6 short posteroventral to ventral setae in distal fourth and with only two short posterodorsal setae in row near distal third of femur; f_2_ elongated and strongly thickened, in proximal third broadest (twice as thick as f_3_ in this place), gradually tapered distally where as wide as f_3_; both f_2_ and f_3_ without specific setae, uniformly densely finely setulose; t_1_ finely setulose but with 1 small dorsal preapical seta (paired with 1 setula); t_2_ (besides usual short setosity) with a long row of 11 or 12 erect posterior setae (those in the middle longest, see Figure 10D) in distal three-fourths, only two very small dorsal setae (one preapical, the other in distal fifth) and 1 short but robust ventroapical seta plus 2–3 smaller but thick setae on apex around it; t_3_ very slightly bent (Figure 10A), without dorsopreapical seta, but with 1 short ventroapical (as long as that on f_2_) and one smaller anteroapical seta, otherwise uniformly setulose. Tarsi simple, slender; fore basitarsus with two longer fine setulae ventrobasally, other setulae short and dense; mid and hind basitarsi long (longer than rest of tarsus) and each with a row of thicker posteroventral erect setulae.

***Wing*** (Figure 10A). Relatively broad although less than that of *A. tertiaria*; veins brown; membrane uniformly brown darkened. C attenuated beyond R_4+5_ but reaching to M; Cs_2_ and partly also Cs_3_ (in basal eighth only) with thicker sparse spine-like setulae in addition to fine pilosity. Sc fine, well developed also basally. R_1_ setulose (five setulae recognized), with distinct preapical kink. R_2+3_ long, very slightly sinuate (almost straight), apically slightly upcurved to C. R_4+5_ shallowly but distinctly bent posteriorly, distally subparallel with M. Distal part of M very slighly bent and reaching wing margin. Discal (dm) cell elongated and relatively narrow (particularly basally); cross-vein r-m situated slightly in front of the middle of cell. Distal part of CuA_1_ about 1.3 times as long as dm-cu and almost reaching wing margin; A_1_ long, but ending far from wing margin. Alula probably large and broad (folded in holotype). Wing measurements: length 2.58 mm, width 1.11 mm, Cs_3_:Cs_4_ = 2.06, r-m\dm-cu:dm-cu = 2.78. Haltere unicolourous, brown.

***Abdomen*** (Figure 1B, Figure 10A and Figure 11C). Subcylindrical, elongately pyriform in dorsal view, slightly bent in lateral view (Figure 11C). Preabdominal terga rather densely and short setose, with longest setae at posterior margins; setae on T1 proximally and some in the middle of T2 erected. T1 and T2 brown, dorsally somewhat paler, T3–T5 blackish brown. T1−T2 separation discernible. T1–T3 only slightly bent laterally. T1 shortest, T2–T4 subequal in length, T5 somewhat shorter than T4. Preabdominal sterna not visible due to milky coating of ventral side of abdomen, only some setae of S5 discernible. Postabdomen (Figure 11C) with all sclerites probably well developed and asymmetrical. Dorsal pregenital synsclerite T6 + S8 large, slightly asymmetrical, blackish brown and densely short setose. S6 and S7 not visible, both hidden under T6 + S8 and milky coating.

**Figure 11 insects-12-01123-f011:**
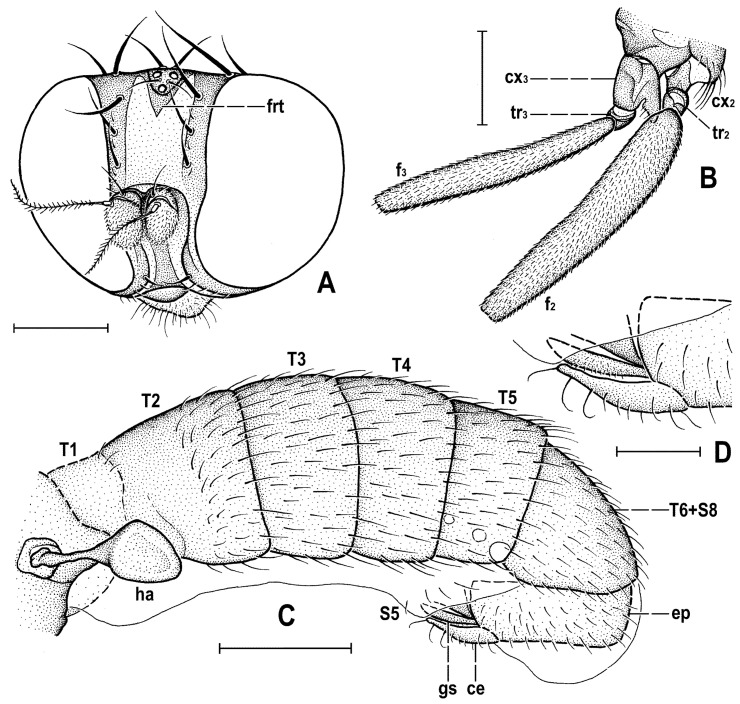
*Acartophthalmites crassipes* sp. nov., male holotype and female paratype (no. 1175-1b): (**A**) holotype, head, frontally; (**B**) paratype, basal parts of right mid and hind legs, laterally; (**C**) holotype, abdomen, left laterally; (**D**) holotype, distal end of genitalia, enlarged. Scales: 0.3 mm (**A**,**C**), 0.5 mm (**B**), 0.1 mm (**D**). For abbreviations see text (p. 3).

***Genitalia***. Epandrium (Figure 11C,D) obscured by milky coating but apparently blackish brown, short and high, with annal fissure directed ventrally, entirely shortly uniformly setulose. Two slender appendages (see Figure 11C,D) are visible at ventral margin of epandrium: posterior, pale brown and longer cercus bearing some curved fine setae along posterior margin, and anterior (closely adjacent) shorter, darker and finely pilose sclerite representing gonostylus or its posterior part.

Female. Similar to male unless mentioned otherwise. Larger, total body length ca 3.8–4.1 mm (Figure 12A,B,E and Figure 13A). ***Head***. Eye (of paratype no. 1175-1b) with posteroventral margin more excavated (cf. Figure 12A) and hence with longest diameter 1.6–1.7 times as long as shortest. Palpus with a few small dark setulae in addition to subapical seta. First antennal flagellomere darker (Figure 12D), orange brown but contrasting with darker pedicel. ***Thorax***. Scutum with up to 12 rows of ac microsetae on suture. One paratype (no. 897-3) aberrantly with two long hu (visible on left side of thorax) and one dc microseta in front of anterior dc more enlarged. ***Legs*** with femora as in male (including strongly swollen f_2_) but f_1_ with three posterodorsal setae in short row in middle third of tibia); cx_3_ similarly posteroventrally projecting as in male (see Figure 10B). t_2_ with 10–11 erect setae (Figure 12F) in posterior long row and with distinctly longer ventroapical seta. ***Wing***. C with spine-like setulae only in Cs_2_. R_1_ with five or six setulae in distal half, beyond kink. One paratype (no. 1175-1b) with wing somewhat more elongate and with R_4+5_ less bent (see Figure 12B) than that of holotype male or of other paratype female. Wing measurements: length 2.78–3.18 mm, width 1.29–1.35 mm, Cs_3_:Cs_4_ = 2.15–2.32, r-m\dm-cu:dm-cu = 2.31–2.90. Haltere brown or with slightly paler knob (only in paratype no. 1175-1b).

**Figure 12 insects-12-01123-f012:**
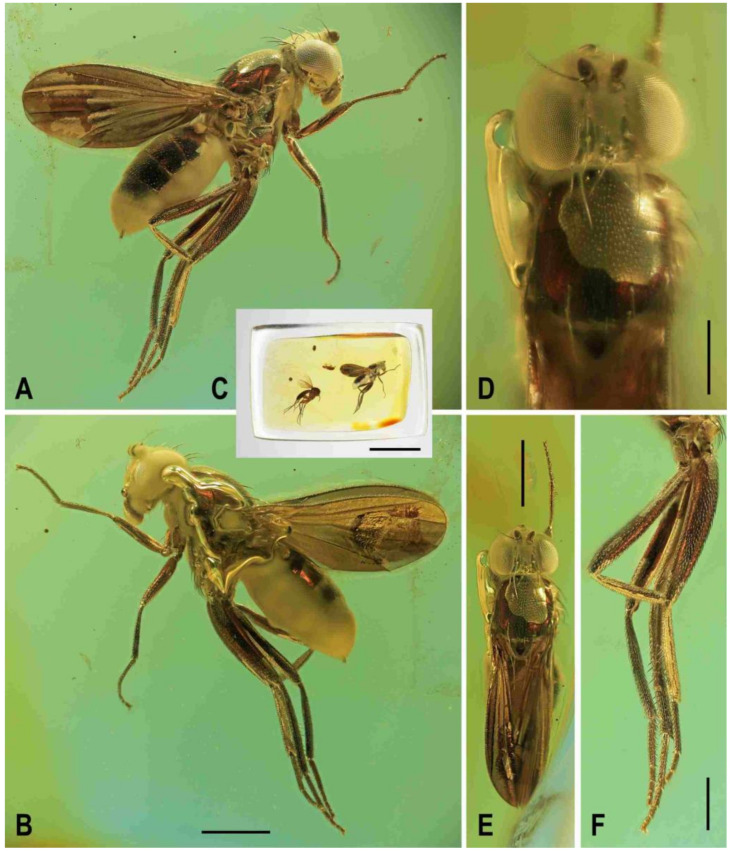
*Acartophthalmites crassipes* sp. nov., female paratype (no. 1175-1b: (**A**) paratype, right laterally; (**B**) ditto, left laterally; (**C**) whole amber sample (preparatum in polyester resin), in situ; (**D**) head and thorax, dorsally; (**E**) paratype, dorsally; (**F**) mid and hind legs, right laterally. Scales: 1 mm (**A**,**B**,**E**), 5 mm (**C**), 0.5 mm (**D**,**F**). Photos by J. Roháček.

***Abdomen*** (only in paratype no. 1175-1b discernible, Figure 12A) less broad than that of *A. rugosus* or *A. tertiaria*, elongately subovoid in dorsal view. Preabdominal terga largely blackish brown, slightly shorter but more transverse than in male but similarly setose. T1 brown, narrow, and indistinctly (laterally) separate from T2; T3–T5 blackish brown; T2 anteriorly brown, posteriorly blackish brown, widened posteriorly, longer but slightly narrower than T3; T3 and T4 probably subequal in length and width, strongly transverse; T5 also transverse, slightly shorter and narrower than T4 and somewhat tapered posteriorly. Ventral side of abdomen obscured by heavy layer of milky coating and hence only S2 partly visible, paler brown than T2, rectangular, only slightly wider than long, shortly finely setose.

***Postabdomen***. Largely undescribed due to milky coating of end of abdomen, with only cerci projecting. S8 partly visible in ventrocaudal view, small, narrow but probably transverse, pale brown a with a few fine setae. Cerci (Figure 12A) very slender, elongate, brown, each with distinct hair-like setae laterally and apically.

**Figure 13 insects-12-01123-f013:**
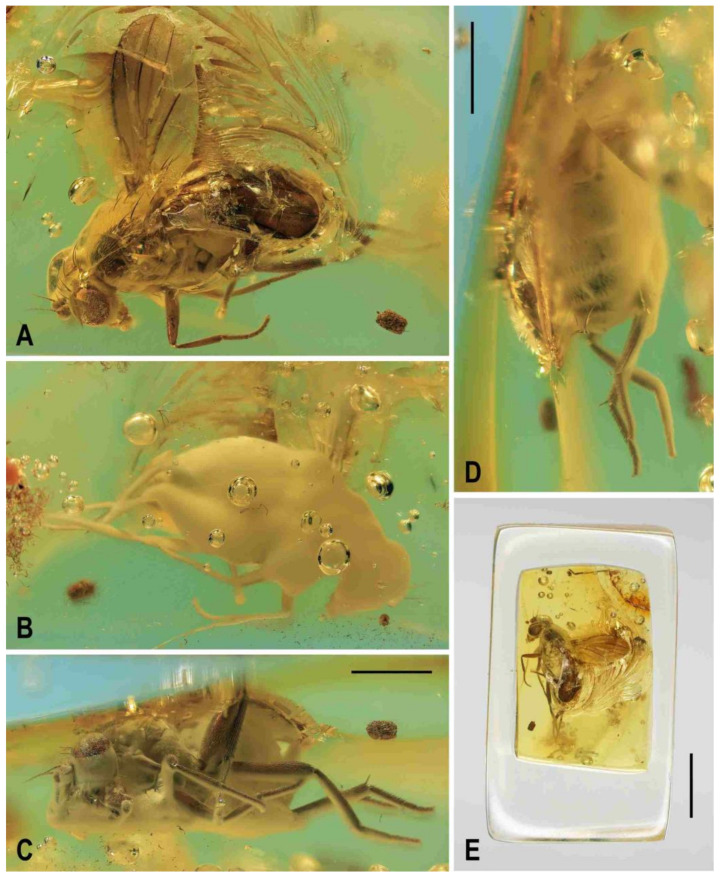
*Acartophthalmites crassipes* sp. nov., female paratype (no. 897-3): (**A**)–paratype, left laterodorsally; (**B**)–ditto, right lateroventrally; (**C**) ditto, subventrally; (**D**) ditto, subdorsally; (**E**)–whole amber sample (preparatum in polyester resin), in situ. Scales: 1 mm (**A**–**D**), 3 mm (**E**). Photos by J. Roháček.

**Comments**. *Acartophthalmites crassipes* sp. nov. is structurally most similar to *A. luridus* sp. nov. (see the key below) and could be its nearest relative (the conspicuously thickened f_2_ could be a synapomorphy). However, it strikingly differs from the latter species by its extremely dark body colouration (including dark fumose wings) and by a longer row of more robust posterior erect setae on t_2_.


(2)*Acartophthalmites luridus* sp. nov. (Figure 14, Figure 15, Figure 16, Figure 17 and Figure 18)


LSID urn:lsid:zoobank.org:act:BDB84612-B143-41E9-9C3F-D1224E503EC0

**Type material**. Holotype male, labelled: “Faszination Bernstein, Christel Hoffeins, Hans Werner Hoffeins” (framed on obverse), ”1818-1 Acartophthalmidae, Acartophthalmites ♂” (handwritten by C. Hoffeins, on reverse), ”Russia: Kaliningrad Region, Baltic Sea coast, Yantarny” (printed) and “Holotypus ♂, Acartophthalmites luridus sp.n., J. Roháček & C. Hoffeins det. 2021” (red label), deposited in CCHH, no. 1818-1 (block-shaped amber piece ca 8.9 × 6 × 5.2 mm, embedded in polyester resin, size 11.9 × 10.5 × 8.3 mm) (Figure 14D); syninclusions: none.

Paratype male, labelled “Index Coll. 30-2, Acartophthalmites tertiaria, sex?, ACARTOPHTHAL, Kartei, leg. et (10/08) det. M. v. Tschirnhaus” (partly printed, rest handwritten by M. v. Tschirnhaus), ”Russia: Kaliningrad Region, Baltic Sea coast, Yantarny, purchased from Andrey Krylov” (printed) and “Paratypus ♂, Acartophthalmites luridus sp.n., J. Roháček & C. Hoffeins det. 2021” (yellow label), deposited in CMTB, no. 30-2 (subovoid flat amber piece ca 16.3 × 10.8 × 3.6 mm, embedded in polyester resin, size 19 × 13.6 × 5.4 mm) (Figure 15C); syninclusions: 1 stellate hair + several rotten plant remnants.

Paratype female, labelled: “Faszination Bernstein, Christel Hoffeins, Hans Werner Hoffeins” (framed on obverse), ”1818-2 Acartophthalmidae, Acartophthalmites ♀” (handwritten by C. Hoffeins, on reverse), ”Russia: Kaliningrad Region, Baltic Sea coast, Yantarny” (printed) and “Paratypus ♀, Acartophthalmites luridus sp.n., J. Roháček & C. Hoffeins det. 2021” (yellow), deposited in CCHH, no. 1818-2 (block-shaped amber piece ca 6.8 × 6.6 × 3.5 mm, embedded in polyester resin, size 9.7 × 9 × 6.5 mm) (Figure 17E); syninclusions: minute rotten plant remnants, one stellate hair.

**Type locality:** Russia: Kaliningrad Region, Baltic Sea coast, Yantarny (formerly Palmnicken). Note: all amber samples with type specimens were purchased from commercial sources in Lithuania and Russia who got the material from Yantarny.

**Etymology:** The name “luridus” (=pale, from Latin, adjective) of the new species is derived from its strikingly pale (ochreous to orange) body colouration.

**Diagnosis.** Pale ochreous to orange-ochreous species with pale ochreous-brown unicolourous wings; frons surface smooth; frontal triangle small and short; orbit glabrous and shining; ors and particularly oc short; orbit with or without additional microsetulae; palpus orange-yellow with black setulae; scutum without contrasting pattern; two dc, anterior dc small to very small; prescutellar ac setae small and distinctly beyond posterior dc; laterobasal sc distinctly shorter (0.6 times as long as) than apical sc; f_2_ strongly thickened and about as long as f_3_; t_2_ with a row of eight or nine relatively short erect posterior setae; t_3_ distinctly bent; apical part of CuA_1_ as long as or shorter than dm-cu.

**Figure 14 insects-12-01123-f014:**
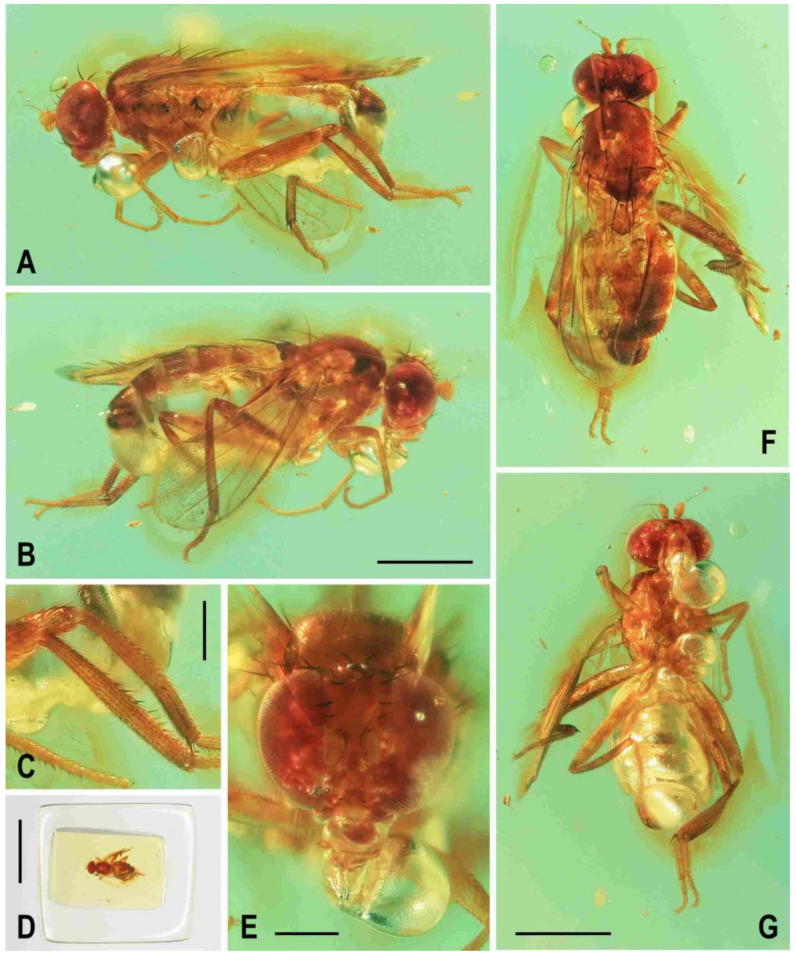
*Acartophthalmites luridus* sp. nov., male holotype: (**A**) holotype, left laterally; (**B**) ditto, right laterally; (**C**) left t_2_ and t_3_, anteriorly; (**D**) whole amber sample (preparatum in polyester resin), in situ; (**E**) head, frontally; (**F**) holotype, dorsally; (**G**) ditto, ventrally. Scales: 1 mm (**A**,**B**,**F**,**G**), 0.3 mm (**C**,**E**), 5 mm (**D**). Photos by J. Roháček.

**Description.** Male. Total body length about 3.5 mm (holotype)–3.8 mm (paratype); general colour pale orange-ochreous or ochreous, including thoracic pleuron which can be darkened dorsally (Figure 14A,B,F,G and Figure 15A,B).

**Head** (Figure 14E, Figure 15D and Figure 16A,B) with occiput dorsally somewhat concave. Head orange (in holotype) to pale ochreous, occiput can be slightly darker. Frons almost unicolourous, narrow but wider than eye in dorsal view, slightly tapering anteriorly, smooth on surface. Orbit well delimited because glabrous and shining in contrast to dull (and somewhat darker) medial part of frons. Frontal triangle dull but slightly paler than vicinity, relatively small and narrow, with anterior corner acute and reaching half of frons; ocellar triangle somewhat darker orange-ochreous and more shining, about half length of frontal triangle and distinctly protruding; ocelli of moderate size (Figure 16B). Frontal lunule small, short and somewhat depressed, concolourous with adjacent frons. Face narrow, very slightly concave; parafacialia indistinctly delimited. Postgena and adjacent part of occiput relatively large. Cephalic chaetotaxy (Figure 16A,B) with most macrosetae shorter than in relatives: pvt longer than vte; vti strong and long, more than twice as long as vte; oc small (as long as foremost ors); three ors, all relatively short but robust and becoming shorter anteriorly, hindmost ors longest and strongest (about as long as pvt); no additional microsetulae on orbit; postocular setulae relatively sparse, in one or two rows (dorsally fewer) at posterior eye margin plus a few additional setulae scattered on adjacent parts of occiput and postgena; postgena with three fine posteroventral setae (hindmost longest) in addition; vi short but distinct, thicker and about twice as long as foremost peristomal setula (vibrissae curved medially, with apices sometimes (paratype) meeting medially); peristomal setulae small and fine (5–7 visible). Eye broadly subovoid (with posterior margin slightly excavated), its longest diameter only about 1.2 times as long as shortest diameter. Eye originally red (remnants of reddish colour preserved). Gena very low, its height only 0.06 times as long as shortest eye diameter. Palpus orange-yellow, small but relatively long (Figure 16B), with distinct black setulae (one longer subapical, 3–4 dorsal and several lateral to ventral shorter setulae). Mouthparts pale ochreous; clypeus darker, ochreous-orange; labellum pale, fleshy and finely pale haired. Antenna entirely ochreous-yellow and relatively small; pedicel externo-laterally with anterior margin slightly convex (mainly dorsally, see Figure 16E), with one longer seta dorsally and 2 or 3 finer setae ventrally besides marginal and submarginal setulae; first flagellomere slightly longer than pedicel, laterally compressed and suboval (distinctly higher than long) in profile, finely whitish pilose (with longer dense cilia on anterior margin). Arista about 2.1 times as long as antenna, ochreous including shortly cylindrical basal segment and its terminal section with relatively short (but longer than in *Clusiomites* spp.) and fine ciliation (Figure 16E).

***Thorax*** (Figure 14A,F, Figure 15A and Figure 16A). Very slightly narrower than head, largely orange (or reddish) ochreous. Scutum somewhat paler laterally and posteriorly, also notopleural area pale ochreous; scutellum and subscutellum more or less distinctly darker than adjacent part of scutum; pleural part of thorax dorsally darker (in holotype with distinct longitudinal stripe) and becoming pale ochreous to orange-ochreous ventrally. Scutellum relatively short (twice wider than long), transversely subtriangular with rounded apex and very slightly convex dorsally; subscutellum distinct, slightly bulging. Thoracic chaetotaxy (Figure 14F and Figure 16A): one hu (strong but shorter than anterior npl); two npl, anterior slightly longer than posterior; one strong sa, longer than anterior npl; two pa, anterior very long and strong (longest thoracic seta), posterior fine and less than one-third of anterior; two dc (both postsutural), both relatively short (in holotype dc setae are distinctly short on right side, particularly right anterior dc is reduced, see Figure 16A); anterior dc situated at level of sa and small to very small, posterior dc robust but only as long as laterobasal sc; one microseta in front of anterior dc sometimes slightly enlarged (as in male paratype); one prescutellar medial ac (situated distinctly beyond level of posterior dc) unusually small (Figure 16A), shorter than posterior pa; scutum covered by microsetae becoming distinctly smaller and denser anteriorly; ac microsetae in about 8–10 rows in front of suture but in fewer posteriorly (only four rows reaching level of posterior dc); two sc, both strong, apical very long but shorter than anterior pa, laterobasal sc distinctly thinner and much shorter (less than 0.6 times as long as) apical sc; one relatively short, fine but distinct upcurved ppl; mesopleuron with one distinct mspl (as long as hu) in posterodorsal corner and scattered (not very numerous) microsetae in posterodorsal half of the sclerite; sternopleuron with one long (as long as mspl) stpl and a several microsetae scattered over all surface, and with one or two longer setae also on ventral corner of sternpleuron.

**Figure 15 insects-12-01123-f015:**
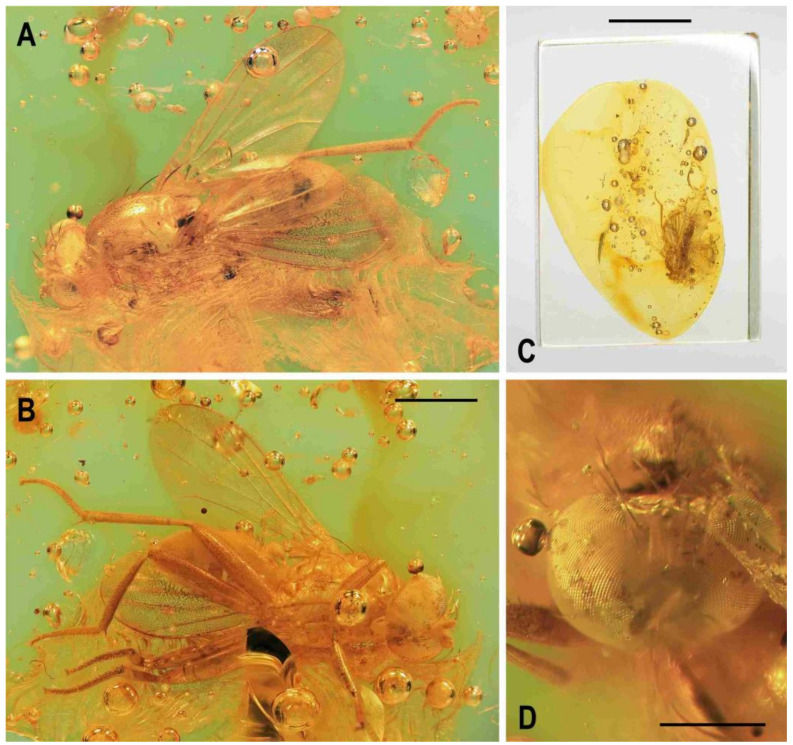
*Acartophthalmites luridus* sp. nov., male paratype: (**A**) paratype, left dorsolaterally; (**B**) ditto, right venntrolaterally; (**C**) whole amber sample (preparatum in polyester resin), in situ; (**D**) head, frontally. Scales: 1 mm (**A**,**B**), 5 mm (**C**), 0.5 mm (**D**). Photos by J. Roháček.

***Legs*** (Figure 14A,C,G). Entirely ochreous to orange-ochreous. Femur and tibia of mid leg about 1.7 times as long as those of fore leg; f_2_ (see Figure 14A,G) markedly thickened and about as long as f_3_; cx_3_ distally (ventrally) projecting and finely setose but this process is apically blunt; f_1_ with a very short row of 4–5 short posteroventral setae in distal fifth and with only two short erect posterodorsal setae in row in middle third of femur; f_2_ elongated and thickened but in proximal third less than twice as thick as f_3_ in this place; both f_2_ and f_3_ uniformly finely setulose; t_1_ finely setulose but with one pair of very small dorsal preapical setae; t_2_ (besides usual short setosity) with a row of eight or nine erect but relatively short posterior setae (those in the middle longest, Figure 14C and Figure 16D) in distal two-thirds, two or three very small dorsal setae (one preapical, others in distal fifth) and one longer and robust ventroapical seta (Figure 16C,D) plus a series of smaller but thick setae (one anterior longer) on apex around it; t_3_ distinctly bent (Figure 14C and Figure 16D), without dorsopreapical seta, but with one short ventroapical (much shorter than that on t_2_), one smaller anteroapical seta and a few thicker setulae on apex. Tarsi simple, slender; fore basitarsus with two longer fine setulae ventrobasally; mid and hind basitarsi long (longer than rest of tarsus) and each with a row of thicker posteroventral erect setulae. 

***Wing*** (Figure 14B and Figure 15B). Moderately broad, surely distinctly narrower than that of A. tertiaria; veins ochreous to yellow; membrane uniformly pale ochreous-brown. C finely setulose but Cs_2_ and partly also Cs_3_ (in basal third) with thicker spine-like setulae in addition. Sc fine but complete, running close R_1_ basally. R_1_ distally setulose (only three setulae present), with distinct preapical kink (in paratype markedly projecting). R_2+3_ very slightly sinuate (basally almost straight), apically slightly upcurved to C and ending distinctly far from wing apex. R_4+5_ slightly but distinctly bent posteriorly, distally almost straight. Distal part of M very almost straight, parallel with R_4+5_ and reaching wing margin. Discal (dm) cell elongated and relatively narrow; cross-vein r-m situated at about proximal two-fifths of cell. Distal part of CuA_1_ rather short, 0.9–1.0 times as long as dm-cu and almost reaching wing margin; A_1_ long, but ending far from wing margin. Alula large and broad (but less than in *A. rugosus)*. Wing measurements: length 2.58 mm (holotypype)–3.02 mm (paratype), width 1.09 mm (holotype)–1.27 mm (paratype), Cs_3_:Cs_4_ = 1.89 (holotype)–2.14 (paratype), r-m\dm-cu:dm-cu = 2.57 (holotype)–2.82 (paratype). Haltere pale ochreous, stem somewhat darker.

**Figure 16 insects-12-01123-f016:**
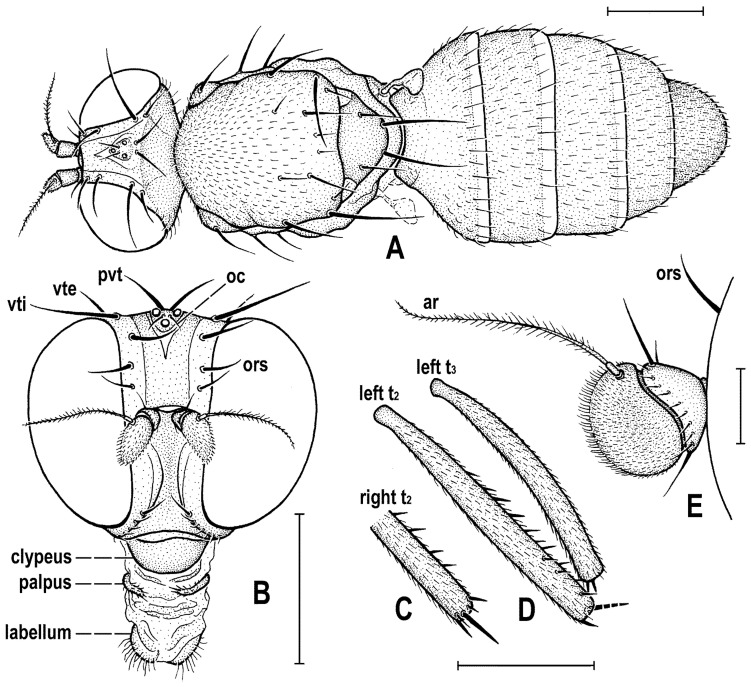
*Acartophthalmites luridus* sp. nov., male holotype: (**A**) whole body without extremities, dorsally; (**B**) head, frontally; (**C**) distal part of right t_2_, ventrally; (**D**) left t_2_ and t_3_, anteriorly; (**E**) left antenna, laterally. Scales: 0.5 mm (**A**–**D**), 0.1 mm (**E**). For abbreviations see text (p. 3).

***Abdomen*** (Figure 14B,F,G and Figure 16A). Subcylindrical, elongately pyriform in dorsal view (broadest in anterior third, Figure 16A), very slightly bent in lateral view (Figure 14B). All preabdominal terga ochreous-orange or pale brown and rather densely but short setose; setae on T1 proximally and some in the middle of T2 more or less erected. T1 and T2 somewhat paler than T3–T5. T2–T4 with distinct dirty whitish posterior marginal band; T5 only very narrowly pale margined posteriorly. T1−T2 separation poorly visible. T1 shortest and narrow, T2–T4 subequal in length, T5 shorter than T4. Preabdominal sterna not visible in holotype due to large air bubble on ventral side of abdomen, but S3 and S4 are partly discernible on paratype (Figure 15B); both S3 and S4 pale ochreous, subequal in length, slightly transverse and markedly narrower than adjacent terga (pleural membrane large), very finely setulose. Note: Most of abdomen of paratype is covered by milky coating but its general shape demonstrates that it is male. Postabdomen (of holotype) with only synsclerite T6 + S8 visible (see Figure 14A,B,F and Figure 16A), the latter longer than T5, relatively narrow (almost as long as wide) and slightly asymmetrical, larger on left side, brown (darker than T5) and densely short setose.

***Genitalia***. Largely obscured by voluminous air bubble (Figure 14A,B,G) in holotype, invisible in paratype. Epandrium (of holotype) only partly visible on right side (Figure 14B), brown, short and high, with annal fissure directed ventrally, entirely shortly uniformly setulose (setosity distinctly shorter and finer that that of T6 + S8). 

**Figure 17 insects-12-01123-f017:**
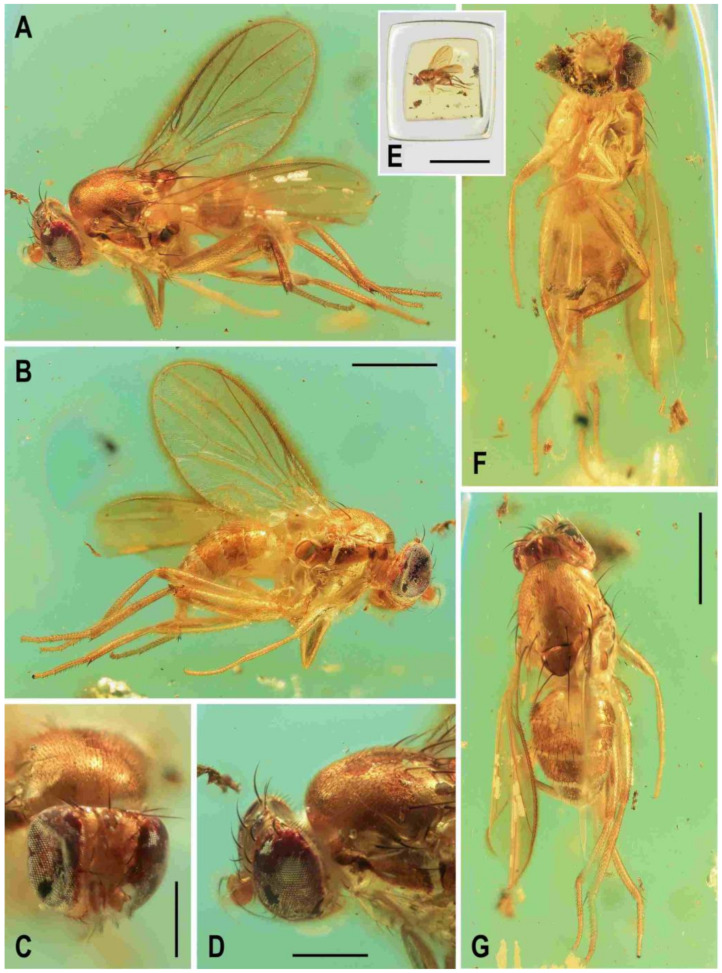
*Acartophthalmites luridus* sp. nov., female paratype: (**A**) paratype, left sublaterally; (**B**) ditto, right sublaterally; (**C**) head, frontally; (**D**) head and anterior part of thorax, left sublaterally; (**E**) whole amber sample (preparatum in polyester resin), in situ; (**F**) paratype, ventrally; (**G**) ditto, dorsally. Scales: 1 mm (**A**,**B**,**F**,**G**), 0.5 mm (**C**,**D**), 5 mm (**E**). Photos by J. Roháček.

Female. Similar to male (holotype in particular) unless mentioned otherwise. Total body length ca 3.2 mm. ***Head*** (Figure 17C,D). Orbit with one or two additional microsetulae between ors; postocular setulae somewhat longer. Eye with red colouration well preserved (Figure 17C) and somewhat higher, with longest diameter about 1.4 times as long as shortest. ***Thorax***. Anterior dc longer than posterior pa (Figure 17A,G) and no enlarged dc microseta in from of it. Scutum with about 10 rows of ac microsetae on suture; prescutellar ac pair small as in male but situated more posteriorly. Pleuron with more distinct (brown) longitudinal dorsal band (Figure 17B) and its ventral part paler. ***Legs***. t_2_ with eight erect setae in posterior long row and long ventroapical seta (Figure 18C); t_3_ less but distinctly bent (Figure 17G) and with ventroapical and anteroapical setae subequal. ***Wing***. Somewhat broader (Figure 17A,B). C with spine-like setulae extended only slightly beyond apex of R_2+3_. R_1_ with 4 setulae in distal half; r-m situated closer middle of dm cell. Wing measurements: length 2.70 mm, width 1.21 mm, Cs_3_:Cs_4_ = 2.10, r-m\dm-cu:dm-cu = 2.66.

***Abdomen*** (Figure 17B,G and Figure 18A–C) distinctly less broad than that of *A. rugosus*, elongately subovoid in dorsal view (Figure 18B). Preabdominal terga (Figure 18A,B) largely pale ochreous, somewhat shorter and wider than in male, all densely short setose. T1 only laterally separate from T2; T2–T4 subequal in length; T2 widest at posterior margin, T3–T5 becoming narrower and somewhat darker posteriorly; T5 distinctly narrower and less transverse than T4. Ventral side of abdomen only partly visible (see Figure 18C). Preabdominal sterna (S2–S5) shortly and finely setose. S1 not discernible; S2–S5 becoming slightly wider posteriorly, but all relatively narrow; S2 and S3 subequal, narrow, but slighly transverse, both pale yellow; S4 and S5 ochreous, both somewhat shorter and wider than S3, S5 widest and shortest (most transverse).

**Figure 18 insects-12-01123-f018:**
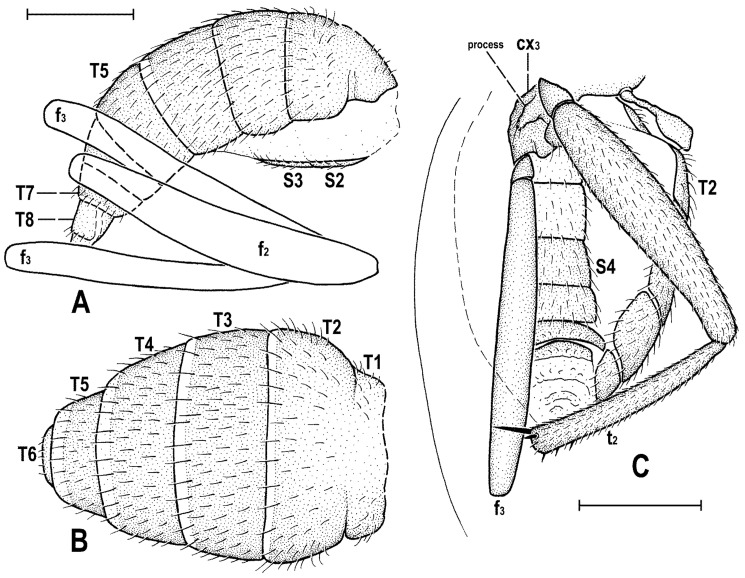
*Acartophthalmites luridus* sp. nov., female paratype: (**A**) abdomen laterally; (**B**) ditto, dorsally (partly reconstructed); (**C**) abdomen and parts of mid and hind left legs, ventrally. Scales: 0.5 mm. For abbreviations see text (p. 3).

***Postabdomen*** (Figure 18A–C). T6 only half length of T5 but slightly narrower; T7 as long as T6 but distinctly narrower, both with sparse fine setae. T8 small and narrow, about as long as wide, with some setulae at posterior margin (cf. Figure 18A); T10 not discernible, probably minute (setosity not visible). Both S6 and S7 darker than S5 and sparsely finely setose; S6 obviously wider and shorter than S5, trasverse; S7 as long as S6 but narrower, both with a few small setae at posterior margin (Figure 18C); S8 and S10 not visible, obscured by dirt. Cerci poorly visible (in caudal view only), slender, elongate, ochreous, apparently setose but setae indistinct.

**Comments**. As noted above, *Acartophthalmites luridus* sp. nov. is probably most closely allied to *A. crassipes* sp. nov. It can be readily recognized from all described Clusiomitidae by its pale body, distinctly bent t_3_ and very small oc setae.


(3)*Acartophthalmites rugosus* sp. nov. (Figure 19, Figure 20, Figure 21, Figure 22, Figure 23 and Figure 24)


LSID urn:lsid:zoobank.org:act:19260A6B-BC45-4F33-B247-8760C8F3F067

**Type material.** Holotype male, labelled: “Faszination Bernstein, Christel Hoffeins, Hans Werner Hoffeins” (framed on obverse), ”1827-15 Diptera: Acartophthalm. Acartophtalmites ♂” (handwritten by C. Hoffeins, on reverse), ”Baltic amber 40 mln years old 46., 0.1 g, 7 × 7 × 3, species Acalyptrata, 83.8, rmvveta, www.ambertreasure4u.com” (printed, partly handwritten), ”Autoclave” (green label), ”Russia: Kaliningrad Region, Baltic Sea coast, Yantarny” (printed) and “Holotypus ♂, Acartophthalmites rugosus sp.n., J. Roháček & C. Hoffeins det. 2021” (red label), deposited in CCHH, no. 1827-15 (circular amber piece 7 × 7 × 3 mm, embedded in polyester resin, size 14.1 × 11.2 × 4.9 mm) (Figure 19E); syninclusions: Sciaridae, one female. The specimen has all tarsi of mid and hind legs partly or wholly cut off (Figure 19A,B). Amber with holotype was treated in autoclave but without any serious modifications (Hoffeins 2012).

Paratype male, labelled “Acartophthalmidae, Acartophtalmites, nr. 2000/52, Coll. Kernegger, ♂, 18. 02. 2015” (pencil handwritten by F. Kernegger), ”Russia: Kaliningrad Region, Baltic Sea coast, Yantarny” (printed) and “Paratypus ♂, Acartophthalmites rugosus sp.n., J. Roháček & C. Hoffeins det. 2021” (yellow label), deposited in CFKH (block-shaped amber piece, ca 9.3 × 7 × 6.2 mm, embedded in polyester resin, size 12.1 × 9.4 × 8.9 mm) (Figure 22A); syninclusions: stellate hairs, dirt.

Paratype female, labelled “Faszination Bernstein, Christel Hoffeins, Hans Werner Hoffeins” (framed on obverse), “1821-1 Diptera: Acalyptratae, Acartophthalmidae, Acartophtalmites ♀” (handwritten by C. Hoffeins, on reverse), “Muscoid 14, 1,3 g, 85.1, 25 × 14 × 5” (obverse, handwritten), “60” (reverse, handwritten), “Russia: Kaliningrad Region, Baltic Sea coast, Yantarny” (printed) and “Paratypus ♀, Acartophthalmites rugosus sp.n., J. Roháček & C. Hoffeins det. 2021” (yellow label), deposited in CCHH, no. 1821-1 (rectangular amber piece 8.6. × 4.6–6.1 × 3.4 mm, cut off the original amber stone) (Figure 24E); syninclusions: small rotten plant remnants (in two remaining parts of the original amber sample)].

**Type locality**: Russia: Kaliningrad Region, Baltic Sea coast, Yantarny (formerly Palmnicken). Note: all amber samples with type specimens were purchased from commercial sources in Lithuania who got the material from Yantarny.

**Etymology:** The species named “rugosus” (=wrinkled, from Latin, adjective) because of its frons distinctly wrinkled (striated) along frontal triangle.

**Diagnosis.** Dark species with unicolourous brown wings; frons dark brown, rugose, coarsely striated along frontal triangle, the latter very narrow and acutely projecting; orbit without additional microsetulae; scutum without contrasting pattern; two dc; prescutellar ac pair situated slightly beyond level of posterior dc; laterobasal sc almost as long as apical sc; t_2_ with a row of 7–8 erect posterior setae; apical part of CuA_1_ about as long as dm-cu.

**Figure 19 insects-12-01123-f019:**
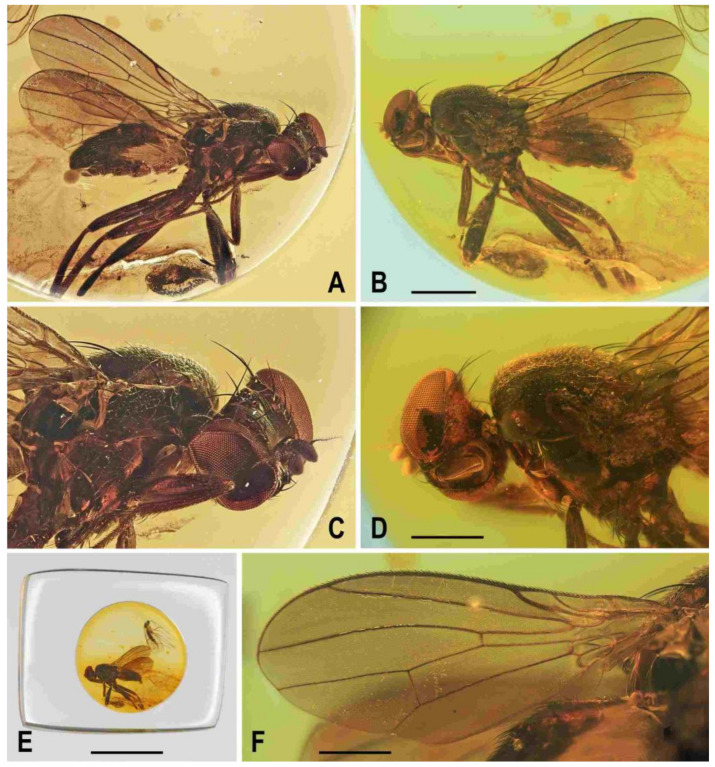
*Acartophthalmites rugosus* sp. nov., male holotype: (**A**) holotype, right laterally; (**B**) ditto, left laterally; (**C**) holotype, head and thorax, right laterally; (**D**) ditto, left laterally; (**E**) whole amber sample (preparatum in polyester resin), in situ; (**F**) right wing, ventral side. Scales: 1 mm (**A**,**B**), 0.5 mm (**B**,**D**,**F**), 5 mm (**E**). Photos by M. Veta (**A**,**C**) and J. Roháček (others).

**Description.** Male. Total body length about 3.3 mm (paratype)–3.8 mm (holotype); general colour blackish brown to brown, with thoracic pleuron somewhat bicolourous (Figure 19A,B and Figure 22 B,C,F).

***Head*** (Figure 19C,D, Figure 20A and Figure 22D) distinctly higher than long; occiput dorsally concave. Head brown to dark brown; occiput darkest; frons largely dark brown to brown, only its anterior sixth pale brown; face, gena and postgena brown. Frons wider than eye in dorsal view, slightly tapering anteriorly, and with distinctive striation (4 striae on each side of frontal triangle, see Figure 19C and Figure 20A) reaching to anterior sixth of frons. Orbit delimited by outernmost stria, darker posteriorly, paler anteriory, dull. Frontal triangle small and narrow, with anterior corner very acute and reaching half of frons; ocellar triangle rather large (two-thirds of frontal triangle) but slightly protruding; ocelli relatively large. Frontal lunule well developed, pale brown. Postgena and adjacent part of occiput relatively large due to shape of eye (see below). Cephalic chaetotaxy (Figure 20A): pvt strong, as long as vte; vti very strong (longest cephalic seta), almost twice as long as vte; oc fine but almost as long as pvt, inserted closely between ocelli; three ors becoming shorter anteriorly, middle ors almost as long as hindmost ors, the latter longest and strongest (slightly longer than vte); no microsetulae on frons; postocular setulae in two rows dorsally surrounding posterior eye margin, and more numerous additional setulae scattered on adjacent lateral parts of occiput and postgena; postgena with two subequal and rather short posteroventral setae in addition; vi relatively short but distinct, much thicker and more than twice as long as foremost peristomal setula (vibrissae curved medially with apices crossed); peristomal setulae small and fine (six or seven visible). Eye large, covering most of head in profile, subovoid (with posterior margin somewhat excavated so that postgena expanded), its longest diameter about 1.45 times as long as shortest diameter. Gena low, its height about 0.1 times as long as shortest eye diameter. Palpus pale brown, ochreous, small and short, with only microsetae on apex visible. Mouthparts ochreous but clypeus dark brown; labellum moderate and fleshy, densely, finely, pale and relatively long-haired. Antenna slightly bent ventrally and relatively small (Figure 19C and Figure 20A), scape and pedicel brown; pedicel externo-laterally with anterior margin very slightly convex, with one stronger erect seta dorsally and two or three finer setae ventrally besides marginal and submarginal setulae; first flagellomere pale brown or ochreous, longer than pedicel, laterally compressed, in profile suboval. Arista relatively short, about 2.1 times as long as antenna, with suboval ochreous basal segment and darker brown terminal section being moderately long-ciliate (Figure 20A).

**Figure 20 insects-12-01123-f020:**
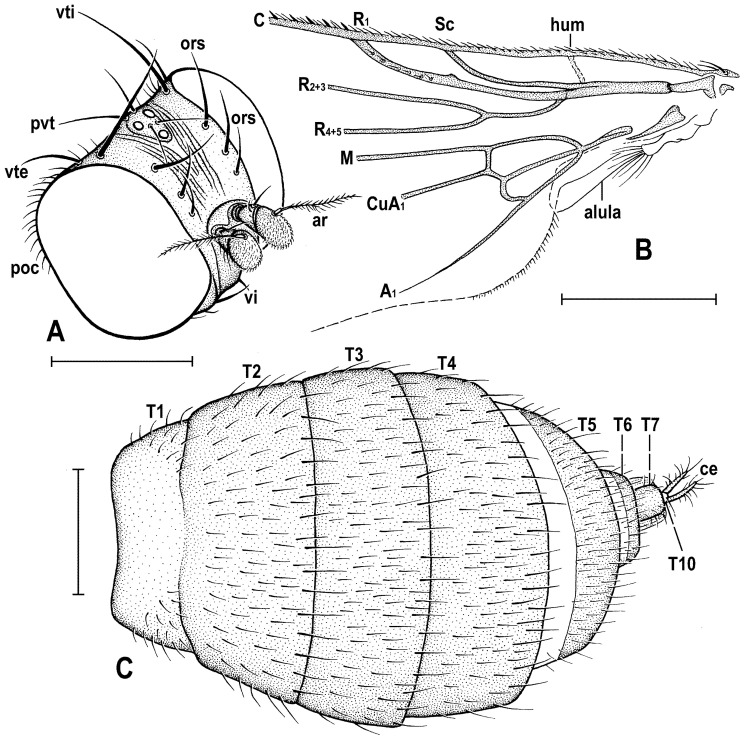
*Acartophthalmites rugosus* sp. nov., male holotype and female paratype: (**A**) holotype, head, right anterolaterally; (**B**) holotype, base of right wing; (**C**) paratype, female abdomen dorsally. Scales: 0.5 mm. For abbreviations see text (p. 3).

***Thorax*** (Figure 19C,D). Somewhat narrower than head, largely dark brown to brown, more bicoloured on pleural part. Scutum dark brown, not patterned, only slightly paler laterally, hence humeral (postpronotal) callus, notopleural area, lateral area above wing base brown; scutellum and subscutellum blackish brown; pleural part of thorax somewhat bicoloured, with broad dorsal band blackish brown; ventral part of pleuron, comprising ventral portion of mesopleuron and propleuron, entire sternopleuron and hypopleuron brown. Scutellum relatively large, subtriangular with rounded apex, slightly convex dorsally; subscutellum distinct but slightly bulging. Thoracic chaetotaxy (Figure 19C,D): one hu (strong and long); two npl, anterior distinctly longer than posterior (in holotype aberrantly only single, anterior, npl present, see Figure 19C!); one robust sa, longer than anterior npl; two pa, anterior very long and strong, (longest thoracic seta), posterior thinner and only one-third of anterior; two dc, anterior situated slightly behind level of sa and about two-fiths of length of posterior, posterior dc robust but slightly shorter than apical sc; oje or two microsetae in front of anterior dc somewhat enlarged, up to twice as long as other dc microsetae; one prescutellar ac (situated slightly beyond level of posterior dc) subequal in length and thickness to posterior pa; scutum otherwise covered by uniform and relatively dense microsetae; ac microsetae in about 10 rows in front of suture but less posteriorly (only six rows reaching level of posterior dc); two sc, both strong, apical very long but shorter than anterior pa, laterobasal sc somewhat thinner but almost as long as apical sc; one long upcurved ppl; mesopleuron with one distinct mspl in posterodorsal corner and numerous microsetae in posterodorsal half of the sclerite, with two or three microsetae below mspl more or less enlarged; sternopleuron with 1 long (posterior) stpl and a number of microsetae scattered over most of surface, two longer but fine setae also on ventral corner of sternopleuron.

***Legs*** (Figure 19A,B, Figure 22B,C,E and Figure 23A). Brown to dark brown (tarsi lighter, femora darker), relatively long and robust. Femur and tibia of mid and hind leg about 1.5 times as long as those of fore leg, mid femur very slightly longer but markedly thicker than hind femur; cx_3_ distally with an acute posteroventral, finely setose, process; f_1_ with a short row of four short posteroventral to ventral setae in distal fifth and with three or four short posterodorsal setae forming a short row in the middle third of femur; f_2_ elongated but thickened, slightly longer but distinctly thicker than f_3_, gradually tapered distally; both f_2_ and f_3_ uniformly densely setulose; t_1_ also finely setulose but with a pair of short dorsal preapical setae; t_2_ (Figure 22E and Figure 23A) with a row of 7–8 sparse erect posterior setae (those proximally longer) in distal two-thirds to three-fifths, only two very small dorsal setae (one preapical, the other above it) and one longer and robust ventroapical seta plus a whirl of several short thicker setae on apex, one anteroventral of which can be longer; t_3_ very slightly bent, without dorsopreapical seta, but with one distinct ventroapical and one anteroapical seta. Tarsi simple, slender; fore basitarsus with one or two longer setulae ventrobasally; mid and hind basitarsi long (longer than rest of tarsus) and with a row of thicker posteroventral erect setulae. 

***Wing*** (Figure 19F and Figure 20B). Elongated but only moderately narrow; veins brown; entire membrane brown darkened. C attenuated beyond R_4+5_ but reaching to M; C finely setulose but Cs_2_ and partly also Cs_3_ (in basal fourth) with thicker sparse spine-like setulae in addition. Sc distinct, fully developed also basally. R_1_ short, with setulae in distal third (four or six visible) and preapical kink well developed (Figure 20B). R_2+3_ long, very slightly sinuate (almost straight), apically slightly upcurved to C and ending distinctly farther from wing apex than M. R_4+5_ shallowly but distinctly bent posteriorly, distally subparallel with M. Distal part of M almost straight and reaching wing margin. Discal (dm) cell elongated but relatively broad; anterior cross-vein (r-m) situated in about middle of cell. Distal part of CuA_1_ subequal in length to dm-cu and reaching wing margin; A_1_ long, but ending in front of wing margin. Alula large and broad (folded in holotype). Wing measurements: length 2.70–3.10 mm, width 1.07–1.23 mm, Cs_3_:Cs_4_ = 2.30–2.45, r-m\dm-cu:dm-cu = 2.18–2.57. Haltere brown with paler knob.

**Figure 21 insects-12-01123-f021:**
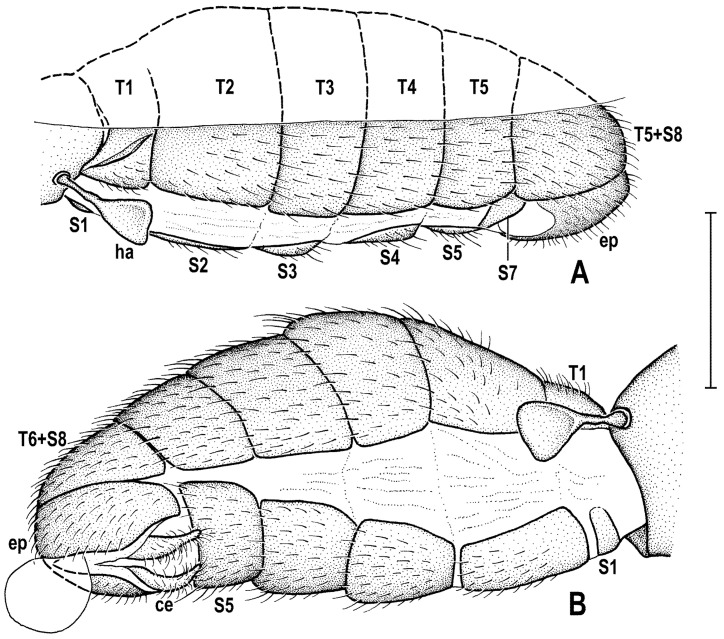
*Acartophthalmites rugosus* sp. nov., male holotype: (**A**) abdomen, left laterally; (**B**) ditto, right laterally. Scale: 0.5 mm. For abbreviations see text (p. 3).

***Abdomen*** (Figure 19A and Figure 21A,B). Subcylindrical, elongate pyriform in dorsal view, widest at T2. All preabdominal terga rather densely and short setose; setae on T1 (proximally) and T2 (in the middle) more or less upright. T1–T3 brown, T1 and T2 dorsally somewhat paler, T4 and T5 blackish brown. T1−T2 separation inconspicuous but laterally discernible. T1–T3 only slightly bent laterally (pleural membrane rather well developed). T1 shortest, T2 longer than T3–T5 (these subequal in length). Preabdominal sterna pale brown to dark brown, all narrower than associated terga and S2–S5 relatively densely but short setose. S1 smallest, short, transverse and bare (Figure 21A); S2 longest, S3 and S4 subequal in length, S5 shorter than S4 but somewhat wider. Postabdomen (Figure 20A,B) with sclerites well developed, asymmetry of postabdominal sterna distinct. T6 + S8 large, long, slightly asymmetrical, blackish brown and densely short setose. S6 and S7 strongly asymmetrical and situated left laterally (S7) to lateroventrally (S6); setosity of S6 and S7 not discernible.

***Genitalia***. Epandrium (Figure 21A,B) blackish brown, relatively short and high, moderately broad, with annal fissure directed ventrally, all shortly uniformly setulose. Annal fissure (Figure 21B) very narrow, particularly dorsally tapered to almost pointed. Cerci distinct, separate, rather symmetrical (Figure 21B) and inserted below annal fissure; each cercus pale brown, slender and distally narrowed, with entire surface finely long-haired. Gonostyli not visible, hidden in genital pouch.

**Figure 22 insects-12-01123-f022:**
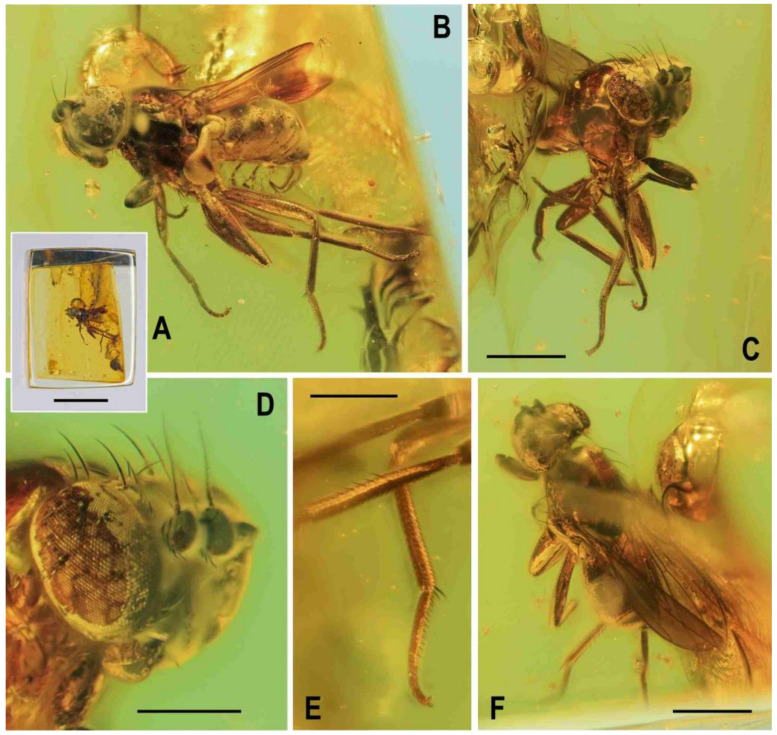
*Acartophthalmites rugosus* sp. nov., male paratype: (**A**) whole amber sample (preparatum in polyester resin), in situ; (**B**) paratype, right anterolaterally; (**C**) ditto, left anterolaterally; (**D**) paratype, head, left anterolaterally; (**E**) legs, subventrally; (**F**)–paratype, subdorsally. Scales: 5 mm (**A**), 1 mm (**B**,**C**,**F**), 0.5 mm (**D**,**E**). Photos by J. Roháček.

**Figure 23 insects-12-01123-f023:**
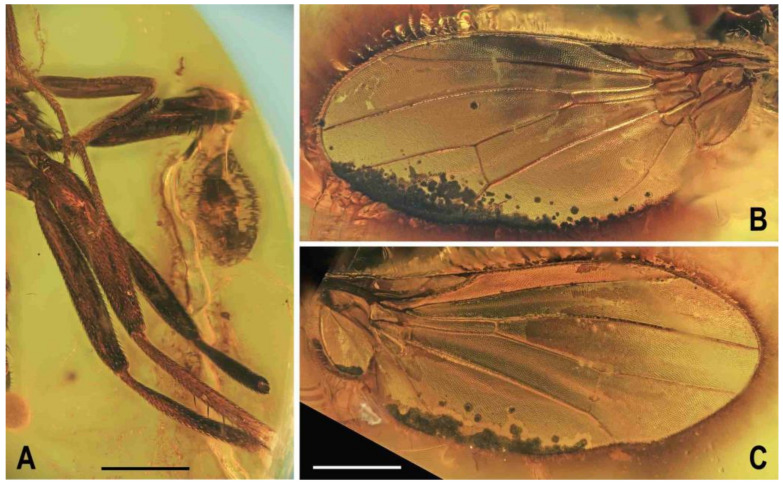
*Acartophthalmites rugosus* sp. nov., male holotype and female paratype: (**A**) holotype, legs, right laterally; (**B**) paratype, left wing, dorsal side; (**C**)–ditto, right wing, dorsal side. Scales: 0.5 mm. Photos by J. Roháček.

Female. Similar to male unless mentioned otherwise (Figure 24A–C). Larger, total body length ca 4.3 mm. ***Head***. Eye with longest diameter 1.5 times as long as shortest. Smallest genal height 0.08 times as long as shortest eye diameter. Gena with seven or eight peristomal setulae. Palpus short, with 1 small but dark subapical seta (not visible in other specimens examined). Arista relatively long-haired. ***Thorax***. Some thoracic setae unnaturally upright (sc in particular, Figure 24B,D) which is an artifact caused probably by movement of amber resin. Pleuron (Figure 21A,C) less distinctly bicolourous, with dark (blackish brown) dorsal half gradually changing to ventral (lighter) brown part. ***Legs*** with femora as in male (tibiae only partly visible) and cx_3_ similarly projecting into posteroventral process. ***Wing***. Membrane seemingly spotted (Figure 23B,C) but this pattern is surely an artefact (caused by air microlayers) because strikingly different on left and right wing. R_1_ with six setulae in distal half, beyond kink. Alula large and very broad (Figure 23B,C). Wing measurements: length 3.49 mm, width 1.41 mm, Cs_3_:Cs_4_ = 2.09, r-m\dm-cu:dm-cu = 2.58.

**Figure 24 insects-12-01123-f024:**
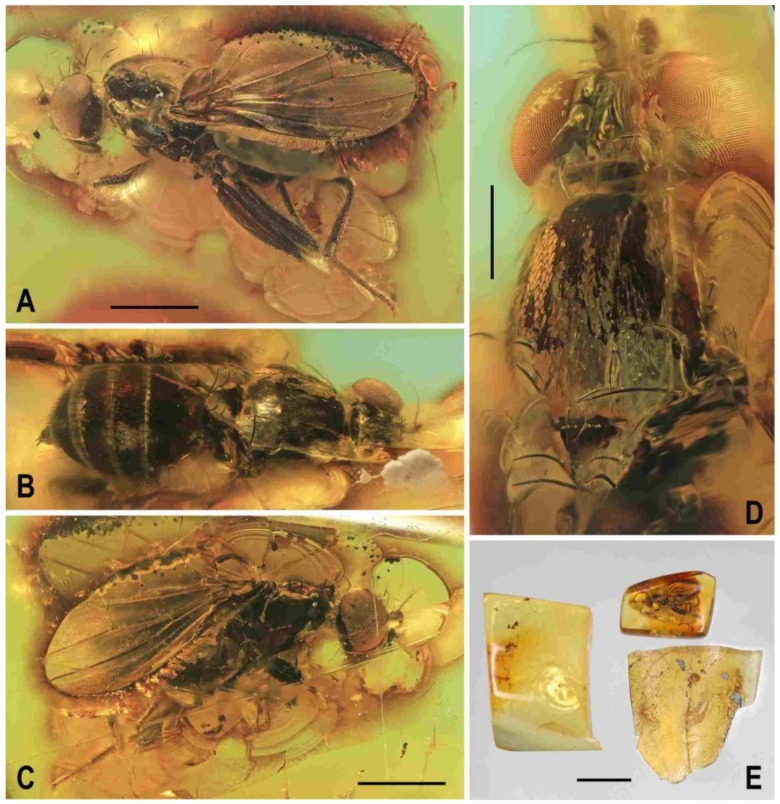
*Acartophthalmites rugosus* sp. nov., female paratype: (**A**) paratype, left laterally; (**B**) ditto, dorsally; (**C**) ditto, right laterally; (**D**) paratype, head and thorax, dorsally; (**E**) amber sample, cut in pieces; specimen with paratype at top right. Scales: 1 mm (**A**–**C**), 0.5 mm (**D**), 5 mm (**E**). Photos by J. Roháček.

***Abdomen*** broad, subovoid in dorsal view (Figure 20C and Figure 24B). Preabdominal terga slightly shorter but much more transverse than in male, all densely short setose, with longest setae at posterior margins. Ventral side of abdomen obscured by air and milky coating and hence sterna not discernible. T1 brown, narrow, and at least laterally separate from T2; T3–T5 blackish brown; T2 strongly widened posteriorly, longer but narrower than T3; T3 and T4 subequal in length and width, very transverse; T5 very transverse, somewhat shorter and anteriorly narrower than T4 but distinctly tapered posteriorly, with posterior corners rounded. 

***Postabdomen***. T6 partly visible (Figure 20C), dark brown, short and transverse, only about one-thirds of width of T5; T7 also dark brown, but small, roughly square-shaped but with posterior corners rounded, laterally surrounded by distinct pleural membrane; T8 indistinct (probably hidden under T7) but T10 (supraanal plate) visible, very small, rounded subtriangular and bearing a pair of long fine setae (only one seta visible). Cerci (Figure 20C) well discernible, long and slender, brown, with distinct lateral and apical hair-like setosity.

**Comments**. *Acartophthalmites rugosus* sp. nov. is particularly well characterized by its longitudinal striation along the frontal triangle and unusually long laterobasal sc (as long as apical sc). It also differs from the other, similarly dark species (*A. crassipes* sp. nov.), by less numerous setae (at most eight) in the posterior row on t_2_ and the terminal part of CuA_1_ not longer than dm-cu. Moreover, the very broad female abdomen of *A. rugosus* is distinctive, most resembling that of *A. tertiaria*, which, however, has the wing broad (see below).


(4)*Acartophthalmites tertiaria* Hennig, 1965 (Figure 25 and Figure 26)


*Acartophthalmites tertiaria* Hennig 1965: 132 [description, illustr.]; Hennig 1969: 18 [list of specimens; note: description on male refers in fact to *A. willii*, see below]; Roháček 2016: 418 [comparative notes]; Pérez-de la Fuente et al. 2018: 137 [comparative notes].

**Type material.** Holotype female, in amber piece mounted on slide and labelled: “Coll, Dr. Klebs (printed), D65, 109 (handwritten), Coll. Dr. Klebs (printed), aka lipte ra (handwritten)” (left label), “fam. Acartophthami-dae, Acartophthal-mites tertiaria Hennig, ♀ Holotypus (handwritten)” (right label), “B M” (written on amber piece, abbreviation of Bernstein Museum), see Figure 25C. Deposited in GZG, inv. no. GZG.BST.00515.

The specimen is in poor condition (examined by C.H. in 2016): clear amber with scattered fine mineral particles of pyrite or hydropyrite, cuticula of the inclusion partly in decomposition due to this mineral impurity; ventrally obscured partly by a milky coating emulsion (German term: Verlumung). The poor state of preservation was already stated by [1] (p. 132), currently its condition is yet worse (see Figure 25A,B).

**Diagnosis.** Generally blackish brown species with pale brown wings. Frons surface smooth; frontal triangle relatively long; bases of antennae distinctly although not far separate; first flagellomere small, hardly longer than pedicel; arista relatively long ciliate; scutum unicolourous blackish brown; anterior pa not longer than sa or apical sc; 3 dc, middle as long as posterior dc; prescutellar ac well developed and situated rather far from base of scutellum (Figure 26F); laterobasal sc robust but shorter (0.7 times as long as) than apical sc; f_2_ slightly longer and distinctly thicker than f_3_; wing broad (Figure 26I), with unicolourous membrane (Figure 25B); R_4+5_ strongly downcurved in basal half; apical part of CuA_1_ distinctly longer than dm-cu; female abdomen short and broad.

**Redescription.** Based on [1] (pp. 130–133, Figures 166, 167, 171, 172, 176–178, 179A,B,D and 180); in description only characters of the holotype female are used), photos by G. Hundertmark and examination of the holotype by C.H. in GZG.

Male unknown. Female. Total body length ca 4.5 mm; general colour blackish brown (Figure 25A,B). Microtomentose pattern of head, thorax and abdomen undescribed due to poor condition of the inclusion.

**Figure 25 insects-12-01123-f025:**
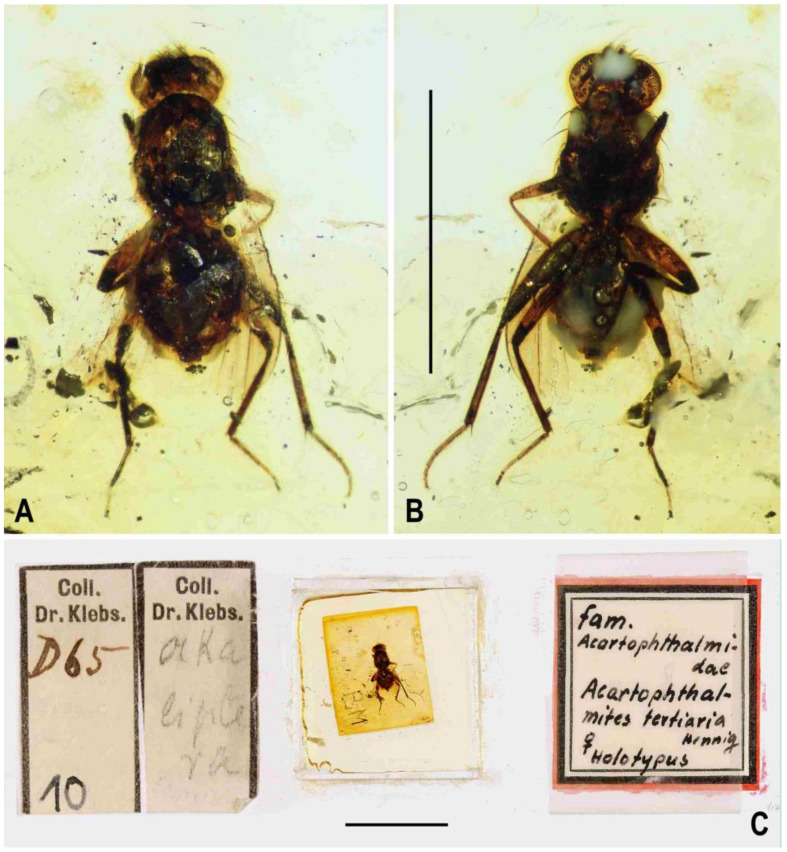
*Acartophthalmites tertiaria* Hennig, female holotype: (**A**) holotype, dorsally; (**B**) ditto, ventrally; (**C**) glass slide with preparatum in situ. Scales: 5 mm (**A**,**B**), 10 mm (**C**). Photos by G. Hundertmark.

***Head*** higher than long (Figure 26A,B). Occiput slightly concave. Frons (cf. Figure 26A) relatively narrow; frontal triangle laterally delimited, not large but relatively long; ocellar triangle small and narrow, with rounded anterior corner; ocelli relatively small. Eye large, vertically subelliptical, with longest diameter oblique, and with posteroventral margin somewhat concave so that ventral part of occiput (and postgena) relatively large (Figure 26B). Parafacialia very narrow. Face evenly convex, without special structures but in profile more protruding than the rather narrow clypeus which is curved downward. Face only very shallowly depressed under bases of antennae and these depressions separated by a flat medial keel (carina). Cephalic chaetotaxy (Figure 26A,B): pvt strong (as long as vte), inserted not far from ocelli, arising on (but not below) rounded upper margin of vertex; vti longest cephalic seta; vte exclinate and distinctly shorter and finer than vti; oc smaller and thinner than pvt; three ors, posterior stronger and longer than both anterior ors; ventral margin of gena with three relatively small setae anteriorly, the foremost longest and representing short vi, peristomal setulae (behind vi) visible anteriorly (2) and posteriorly (4) (see Figure 26B); three longer setae on rounded posteroventral corner of postgena; postgena otherwise with scattered setulae; postocular setulae in two or three rows dorsally, ventrally more numerous and scattered on occiput and adjacent postgena (Figure 26B). Gena low (narrow) while postgena dilated due to posteroventral excavation of eye margin (Figure 26B). Antennae with bases well (but not far) separated (Figure 26A); pedicel conical, bearing 1 longer dorsal seta; 1st flagellomere unusually small (Figure 26B), hardly longer than pedicel, rounded but somewhat flattened laterally; arista about 2 times as long as antenna, rather long ciliate (see Figure 26D) and its second segment relatively short.

**Figure 26 insects-12-01123-f026:**
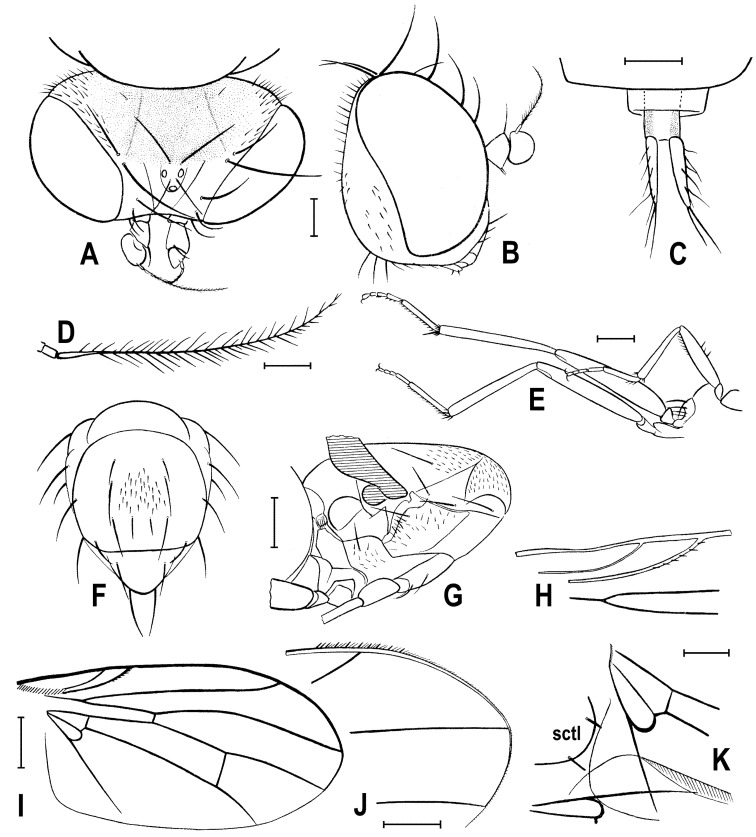
*Acartophthalmites tertiaria* Hennig, female holotype: (**A**) head, dorsofrontally; (**B**) ditto, right laterally; (**C**) end of postabdomen, dorsally; (**D**) antennal arista; (**E**) right legs, ventrally; (**F**) thorax, dorsally; (**G**) thorax right laterally; (**H**) anterobasal venation of wing; (**I**) right wing, dorsally; (**J**) wing apex; (**K**) wing bases. Scales: 0.2 mm (**A**,**B**,**H**,**K**), 0.1 mm (**C**,**D**), 0.5 mm (**F**,**G**,**I**), 0.3 mm (**J**). For abbreviations see text (p. 3. Adapted from [1] (Figures 166, 167, 171, 172, 176–178, 179A,B,D and 180, © Staatliches Museum für Naturkunde Stuttgart).

***Thorax.*** Scutum blackish brown, with transverse suture visible (Figure 26F); also humeral callus (postpronotal lobe) well developed. Scutellum triangular, with rounded posterior corner (Figure 26F). Thoracic chaetotaxy (Figure 26F,G): 1 long hu (as long anterior npl); 2 npl (anterior long, posterior short); 1 sa (as long as anterior pa), 2 pa, anterior robust but not longer than sa or apical sc, posterior short (as long as laterobasal sc); 3 dc, all postsutural, foremost dc shorter but distinct; middle dc as long as hindmost dc (Figure 26F); ac microsetae numerous and somewhat unordered (see Figure 26F,G); 1 distinct pair of prescutellar ac (Figure 26F), situated rather far from base of scutellum at level of posterior dc; 2 sc, apical slightly longer than pa, laterobasal sc relatively long but shorter than (0.7 times as long as) apical sc; 1 distinct upcurved ppl seta (not visible on Figure 26G); one strong mspl and numerous setulae on mesopleuron being denser at posterior margin; one strong (posterior) stpl surrounded by a number of setulae in dorsal half of sternopleuron (see Figure 26G). 

***Legs.*** According to [1] (p. 132) without peculiarities. However, f_2_ and f_3_ long (both markedly longer than f_1_) and f_2_ slightly longer and distinctly thicker than f_3_ (see Figure 25B and Figure 26E); t_3_ straight. According to [1] no (dorsal) preapical setae on any of tibiae. f_1_ with a row of about 7 posteroventral setae in distal two-fifths and with three erect posterodorsal setae in middle third of tibia; t_2_ with a distinct ventroapical seta (Figure 26E) and its apex surrounded by short setae; also apex of t_3_ with a crown of short setae (ventroapical and anteroapical longer than others) judging from Figure 26E. However, it is almost sure that t_2_ in fact also bears a row of (probably short) posterior setae and a few dorsal setae distally (including 1 preapical) as is known in all other congeners. Obviously, these setae were not recognized in the holotype by Hennig [1] due to their weakness (cf. also notes in [20] (137)) or because obscured and hence invisible as are today (examined by C.H. in 2016). Tarsi distinctly paler than tibiae; basitarsi of all legs long, as long as or longer than rest of tarsus.

***Wing*** broad compared to those of all known relatives (Figure 26I), with pale brown membrane and brown to blackish veins, C darkest. C without costal breaks and reaching somewhat beyond apex of R_4+5_ where attenuated (Figure 26J); C with small but distinct spines among fine pilosity last of them reaching slightly beyond apex of R_2+3_ (Figure 26J). Sc well developed, separate and distinct up to junction with C, ending at quite a distance from apex of R_1_ (Figure 26H); R_1_ without preapical kink (based on Hennig‘s illustration) but setulose in distal part (Figure 26H); R_2+3_ not very long, almost straight, with only apex slightly upcurved to C and ending far from wing apex; R_4+5_ strongly downcurved in basal half but straightened terminally and ending at apex of wing. M (beyond cell dm) very slightly bent, subparallel to R_4+5_. Cell dm large and long, with r-m slightly in front of the middle of cell and dm-cu situated relatively far from wing margin; terminal section of CuA1 distinctly longer than dm-cu. Both bm and cup cells are well developed, the latter closed by strongly convex CuA_2_. Anal vein (A_1_) long but apparently not quite reaching wing margin. Alula not visible (Figure 26I), probably broad as in relatives.

***Abdomen*** short and broad, suboval in dorsal view (Figure 25A,B). Preabdominal terga apparently broad, short and hence transverse; sterna undescribed. End of postabdomen narrow and terminating in long, slender and setose cerci (Figure 26C) having longest setae on apex and sublaterally.

**Comments.** Because of its very small first antennal flagellomere, distinctly broad wing and three dc (the middle of which is as long as the posterior one), *Acartophthalmites tertiaria* Hennig, 1965 cannot be confused with any of its congeners. 

However, Hennig [1] also based his description, besides the holotype, on three other females, designated as paratypes. Judging from notes on them in the original description and illustrations [1], they were probably not conspecific with the holotype [19]. Particularly, the Scheele paratype (nr. 1127) differs in a number of characters (larger 1st antennal flagellomere and arista with shorter ciliation, bicolourous pleuron and possibly also scutum, 2 dc, much longer anterior pa etc.) and surely does not belong to *A. tertiaria*. Unfortunately, these three paratypes have not been traced and, therefore, not examined. The paratype (Scheele number 1127) from the Scheele amber collection deposited in GPIH was detected as missing (by C.H. in 2016), similarly as is the second acalyptrate inclusion, *Electroclusiodes meunieri* (Hendel, 1923) (Scheele number 771), mentioned under additional material examined by Hennig [1] (p. 138). The bag with the missing paratype is now containing the GPIH label and a reddish, fragile piece of amber with a damaged female of rhagionid fly in poor condition. Loan and return of the amber material at the GPIH was not documented in any form in the past, hence we can only speculate what may had happened with these inclusions.


(5)*Acartophthalmites willii* Pérez-de la Fuente, Hoffeins & Roháček, 2018 (Figure 27, Figure 28 and Figure 29)


*Acartophthalmites tertiaria*: Hennig, 1969: 18 [misidentification]. 

*Acartophthalmites electrica:* Hennig, 1969: 18–19, Figures 19–21 [error, incorrect subsequent spelling].

*Acartophthalmites willii* Pérez-de la Fuente, Hoffeins & Roháček, 2018: 127.

**Type material.** Holotype male, labelled: “Mus. Comp. Zool. 19475, No. 6545a [the latter number scratched] Haren 115 Coll., Baltic amber”, “Fam. Acartophthalmidae. Acartophthalmites electrica Hennig, ♂”, and “HOLOTYPUS ♂, Acartophthalmites willii sp. n., R. Pérez-de la Fuente, C. Hoffeins & J. Roháček det.” (red label). Deposited in MCZC, no. MCZ-PALE-19475. The amber piece preserving the holotype is encased in an Epoxy resin prism of 18 × 18 × 5 mm and this Epoxy prism is mounted on a glass slide labelled as above. 

Type locality: Baltic sea coast, probably Russia: Samland Peninsula, see [21].

**Diagnosis.** Bicolourous species, with dark brown and ochreous pattern on thorax and wing brownish along anterior margin. Frons surface finely transversely rugose; frontal triangle not visible (not delimited); antennae closely inserted; arista relatively shortly ciliate but ciliation longer (Figure 28) than those of *Clusiomites* species; scutum with contrasting pattern (see Figure 28B); 2 dc setae; prescutellar ac setae well developed, slightly beyond posterior dc; laterobasal sc robust but shorter (0.75 times as long as) than apical sc; f_2_ not longer but distinctly thicker than f_3_, ventrally with single short row of 6 thicker setae; t_2_ with a distal row of erect posterior setae (6 or 7 longer and thicker); wing elongated and darkened along anterior margin; apical part of CuA_1_ short, not longer than dm-cu; male synsclerite T6 + S8 relatively short.

**Description.** Male. Total body length about 3.2 mm (Figure 27A,B); general colour probably bicoloured (Figure 28B), dark brown to light brown and ochreous; legs light brown to ochreous. 

***Head*** (Figure 27D and Figure 28A) distinctly bicoloured; frons mostly light brown; occiput darkest (dark brown); face, gena and postgena ochreous. Frons moderately narrow (slightly wider than eye in dorsal view), slightly tapering anteriorly, largely light brown, with foremost part ochreous and only ocellar triangle dark brown. Frons very finely transversely rugose in texture (Figure 28A). Orbit colouration apparently not distinct from that of frons. Frontal triangle not visible. Ocellar triangle somewhat tubercle-like, somewhat elevated ocelli. Frontal lunule not obvious, probably small. Face (praefrons) dark ochreous, parafacialia and gena ochreous. Postgena and adjacent part of occiput large, expanded, ochreous. Cephalic chaetotaxy (Figure 28A): pvt relatively strong (longer and thicker than oc); vti very strong (longest cephalic seta), almost twice as long as vte; oc relatively weak; three ors becoming shorter anteriorly, the hindmost ors longest and strongest (about as long as vte); no microsetulae on frons medially or in front of ors; postocular setulae dorsally in two irregular rows surrounding posterior eye margin, but there are numerous additional erect setulae scattered on adjacent lateral parts of occiput and postgena (Figure 28B); postgena with two or three (one distinctly longer) posteroventral setae in addition; vi distinct and relatively well developed (Figure 28A), about twice as long as foremost peristomal setula, curved medially; peristomal setulae small and sparse (4 observed). Eye large, subovoid, its longest diameter about 1.35 times as long as shortest diameter. Gena very low, its height about 0.05 times as long as shortest eye diameter. Palpus ochreous, small, with ventropreapical seta longest and a few setulae subdorsoapically. Mouthparts ochreous; labellum large and fleshy, setulation not apparent. Antennae closely inserted, relatively small (Figure 27D and Figure 28A); scape and pedicel light brown; pedicel externo-laterally with somewhat excavated anterior margin, with one stronger erect seta dorsally and two finer setae ventrally in addition to series of marginal and submarginal setulae; 1st flagellomere ochreous, laterally compressed, in profile subcircular with posterior side at least not evidently excavated, probably slightly so. Arista 2.5 times as long as antenna, with elongated and whitish basal segment and darker ochreous terminal section being distinctly (not very shortly) ciliate (Figure 28A).

**Figure 27 insects-12-01123-f027:**
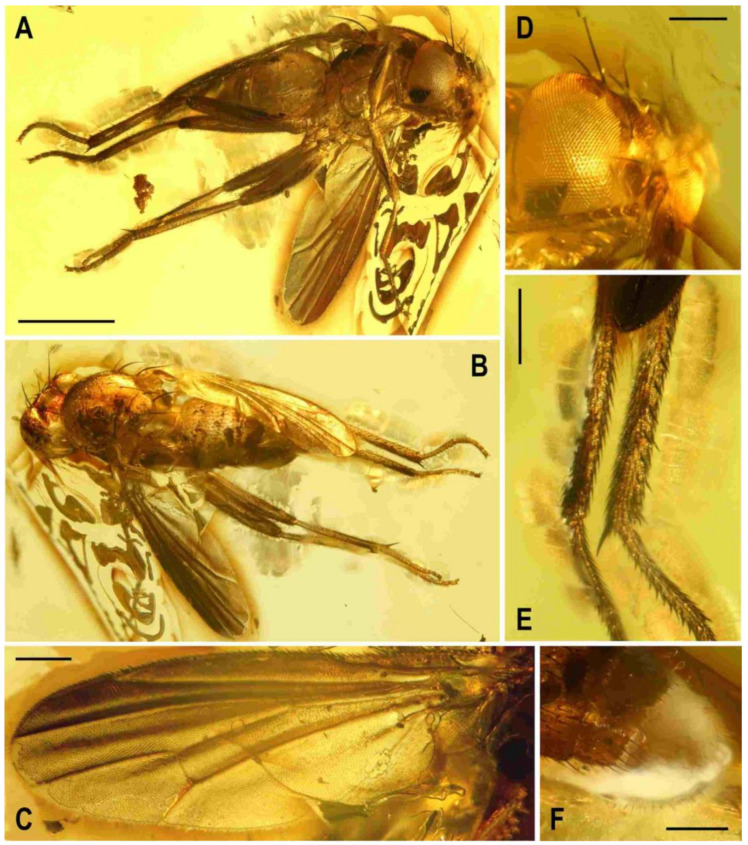
*Acartophthalmites willii* Pérez-de la Fuente, Hoffeins & Roháček, male holotype: (**A**) holotype, right ventrolaterally; (**B**) ditto, left dorsolaterally; (**C**) left wing (slightly narrower then in reality due photographic angle), dorsally; (**D**) head, frontolaterally; (**E**) distal half of right t_2_ and t_3_ and basitarsi, posteriorly; (**F**) apex of abdomen, left dorsolaterally. Scales: 1 mm (**A**,**B**), 0.25 mm (**C**–**F**). Photos by R. Pérez-de la Fuente, adapted from [21] (Figures 1–3, 7, 12 and 13).

***Thorax*** (Figure 27A,B and Figure 28B). Slightly narrower than head, bicolourous, largely pale brown to ochreous, with some parts dark brown. Scutum ochreous, with a longitudinal medial light brown band wider proximally and distally, and two brown spots at the posterior half of the scutum which enclose the sa, pa and the two dc setae (see Figure 28B). Humeral (postpronotal) callus brown and markedly protruding; notopleural area ochreous; scutellum brown and subscutellum light brown; pleural part of thorax distinctly bicoloured: dorsal part of mesopleuron and pteropleuron dark brown; ventral part of mesopleuron and pteropleuron ochreous; metapleuron dark brown; metanotum ochreous; propleuron, sternopleuron and hypopleuron ochreous. Scutellum subtriangular with rounded apex, slightly convex dorsally; subscutellum well developed. Thoracic chaetotaxy (Figure 28B): one hu (strong and long), two npl (anterior slightly longer than posterior), one sa (long and robust but much shorter than anterior pa), two pa (anterior very long and strong, longest thoracic seta; posterior thin and less than half the length of anterior pa); 2 dc, anterior situated slightly behind level of sa and short (about half length of posterior dc), posterior dc robust but clearly shorter than apical sc or anterior pa; enlarged prescutellar ac present immediately after the level of posterior dc, subequal in length and thickness to posterior pa; scutum otherwise covered by uniform and relatively dense microsetae (ca 15 dc microsetae in row in front of anterior dc); ac microsetae arranged in about eight rows in front of suture but less posteriorly, and only 4 rows reaching the level of posterior dc); 2 sc, apical sc strong and very long but shorter than anterior pa, laterobasal dc relatively robust, as long as three-fourths of the apical sc; 1 long ppl; mesopleuron with one distinct mspl in posterodorsal corner and numerous microsetae on most of its surface (except for anterodorsal part); sternopleuron with 1 long stpl and a number of scattered microsetae (anterior part of sclerite covered by a bubble). 

***Legs***. Originally probably all light brown to ochreous, relatively long and slender. Femur, tibia and basitarsus of mid leg about 1.5−1.7 times as long as those of fore leg (Figure 27A), mid tibia only slightly longer than hind tibia (other leg segments of these tibiae subequal in length); cx_3_ with rather acute distoventral setose process directed caudally; f_1_ with a short row of four or five longer posteroventral to ventral setae in distal third and with about four posterodorsal setae forming a row in the middle third of femur; f_2_ elongated, thicker than f_3_ but subequal in length to the latter, not particularly tapered distally, finely densely setulose but ventrally with a single short distal row of six thicker setae; f_3_ without specific setae, uniformly densely finely setulose; t_1_ also uniformly finely setulose but with a pair of small dorsal preapical setae; t_2_ with a row of sparse erect posterior setae (six or seven longer and thicker, Figure 28E and Figure 29A) starting in proximal two-fifths, three short dorsal setae (one in distal third, one at the level of distalmost posterior seta, and one preapical) and one longer and thicker ventroapical seta plus an anteroapical whirl of two or three shorter thicker setae near the latter (Figure 28E) and four smaller, short but thicker setulae also posteroapically (Figure 28D); t_3_ without dorsopreapical seta, but 100 longer thicker ventroapical and 3 shorter thicker setae around the latter (Figure 28C), otherwise uniformly finely setulose. Tarsi simple, slender; fore basitarsus with a few (1 or 2 longer) prolonged fine setulae ventrobasally; mid and hind basitarsi long and with thicker setulae. 

***Wing*** (Figure 27C and Figure 28G). Elongated and relatively narrow; veins light brown; membrane apparently darkened in the anterior third of wing, darkening more restricted to roughly the anterior half of the spaces between C and R_2+3_, and R_2+3_ and R_4+5_ (Figure 28G). C attenuated beyond R_4+5_ but reaching to M; C finely setulose but in Cs_2_ with thicker sparse spine-like setulae in addition. Sc fine but complete (with basal part invisible). R_1_ short and bearing four setulae subapically and preapical kink present at the level of Sc end (Figure 28F). R_2+3_ long, very slightly sinuate, apically somewhat upcurved to C, ending distinctly farther from wing apex than M. R_4+5_ shallowly but distinctly bent posteriorly, distally subparallel with M, ending close to wing apex. Distal part of M very slightly bent and reaching wing margin. Discal (dm) cell relatively elongated; anterior cross-vein (r-m) situated in about middle of discal cell. Distal part of CuA_1_ subequal in length to dm-cu cross-vein and almost reaching wing margin; A_1_ long, ending near wing margin. Alula well developed, large and broad. Wing measurements: length 2.2 mm, width 0.85 mm, Cs_3_:Cs_4_ = 2.05, r-m\dm-cu:dm-cu = 3.24. Haltere ochreous.

**Figure 28 insects-12-01123-f028:**
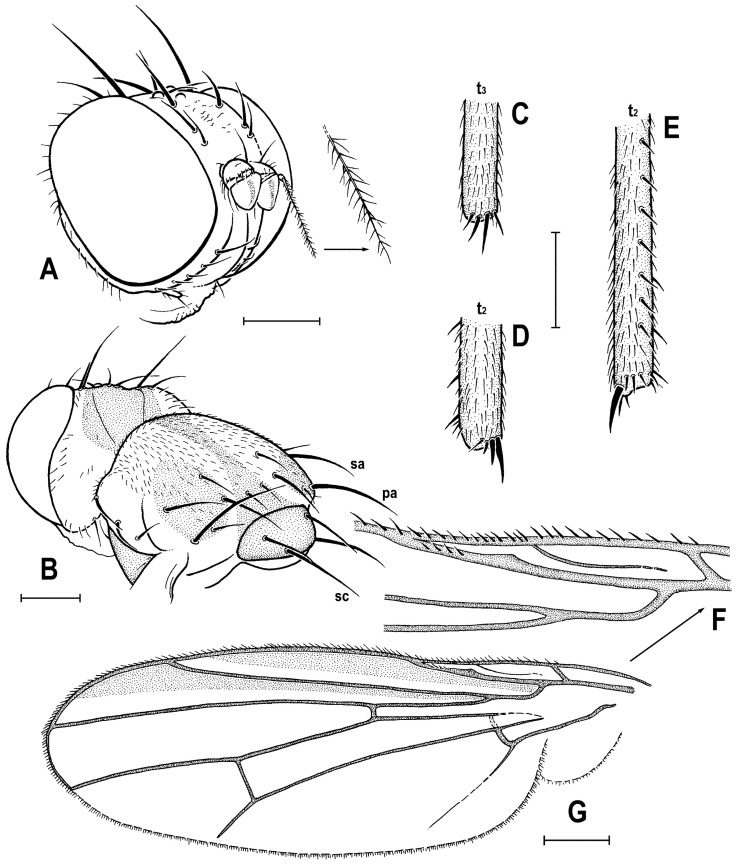
*Acartophthalmites willii* Pérez-de la Fuente, Hoffeins & Roháček, male holotype. (**A**) head, frontolaterally, with distal part of arista enlarged; (**B**) head and thorax, left posterlaterally; (**C**) distal fourth of right t_3_, anteriorly; (**D**) distal fourth of right t_2_, anteriorly; (**E**) distal half of right t_2_, posteriorly; (**F**) anterobasal part of wing left wing; (**G**) left wing, dorsally. Scales: 0.25 mm. For abbreviations see text (p. 3. Adapted from [21] (Figures 15–20).

***Abdomen*** (Figure 27A,B). Pyriform in dorsal view, widest at distal end of T2. All preabdominal terga rather sparsely but distinctly setose, with longest setae (some erected) at posterior and lateral margins. T1–T3 dorsally ochreous, darker laterally, T4 and T5 dark brown. T1−T2 separation not conspicuous. T1–T5 relatively narrow, only slightly bent laterally (pleural membrane rather well developed). Preabdominal sterna ochreous, sparsely and short setose (only S2−S3 visible, both narrow). Postabdomen (Figure 27F and Figure 29B) with sclerites well developed but asymmetry of postabdominal sterna not clearly assessable. Synsclerite T6 + S8 large but not very long, dark brown and densely short setose. S6 and S7 not clearly discernible, but probably asymmetrical and largely situated left laterally. 

**Figure 29 insects-12-01123-f029:**
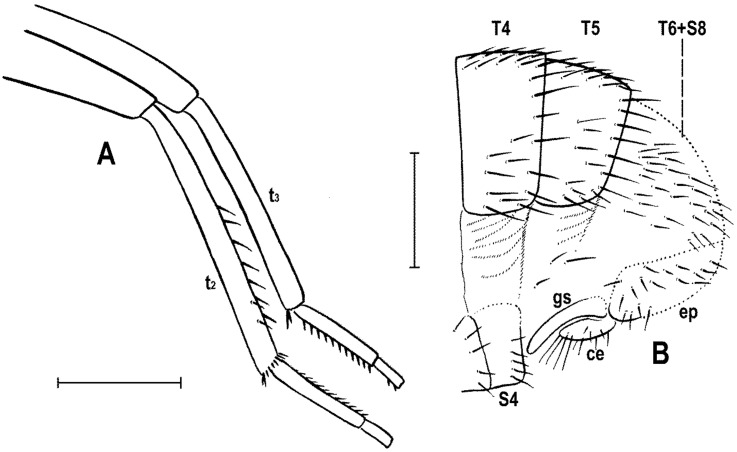
*Acartophthalmites willii* Pérez-de la Fuente, Hoffeins & Roháček, male holotype. (**A**) chaetotaxy of t_2_ and t_3_, anteriorly; (**B**)–apex of abdomen, left laterally. Scales: 0.5 mm (**A**), and 0.3 mm (**B**). For abbreviations see text (p. 3. Adapted from [4] (Figures 20 and 21, © Staatliches Museum für Naturkunde Stuttgart).

***Genitalia***. Epandrium short, width not assessable, uniformly short setose. Cerci barely visible and gonostyli not visible as the sample is currently prepared (but originally depicted in [4] (Figure 20); see Figure 29B). Based on his description and illustration the gonostylus is simple, slender, seemingly bare and slightly bent posteriorly; cercus is elongated, shorter than gonostylus, setose, and with longer setae apically (Figure 29B). Because anteroventral parts of external genitalia are not visible (also to Hennig [4]), we cannot exclude the possibility that he observed and illustrated only the posterior lobe of the left (bilobed) gonostylus. In any case, this lobe is distinctly different from that of *A. crassipes* (the only other species of *Acartophthalmites* where is gonostylus visible) which is slightly bent anteriorly (cf. Figure 11C,D).

Female unknown.

**Comments**. *Acartophthalmites willii* Pérez-de la Fuente, Hoffeins & Roháček, 2018 is the only species of the genus *Acartophthalmites* with a distinctly bicoloured body, thus more resembling in colour species of the genus *Clusiomites*. It also differs from all congeners by its wing pattern (brown darkened along anterior margin and on apex) and f_2_ ventrally distally with a short row of 6 thicker setae. 

The holotype of this species was originally attributed by Hennig [4] (p. 18) as a male of *A. tertiaria*, but it has recently been recognized as a distinctly different species [21]. Hennig [4] was possibly mistaken because one of his paratypes (the Scheele specimen Nr. 1127, see [1] (p. 132) also had a more or less distinct dark pattern on the thorax. We consider this lost paratype (see comments above under *A. tertiaria*) to be different from *A. teriaria*, perhaps being conspecific with *A. willii*.

#### 3.1.4. A Key to Identification of Genera and Species of Clusiomitidae


Body slender (Figure 1A); head as long as high or longer than high (Figure 2F); arista at least 3 times as long as antenna and with short ciliation (Figure 4C,D); R_1_ bare (Figure 3H); f_2_ only slightly thicker than f_3_; male cerci asymmetrical; female abdomen slender (Figure 8A). –genus *Clusiomites* gen. nov. ………...…………… 2
-Body more robust (Figure 1B); head higher than long (Figure 10F); arista about 2 (at most 2.5) times as long as antenna and with longer ciliation (Figure 16E); R_1_ setulose (Figure 20B and Figure 28F); f_2_ distinctly (1.5 × or more) thicker than f_3_; male cerci symmetrical; female abdomen broad (Figure 9A and Figure 24B). –genus *Acartophthalmites* Hennig ………………… 3 Frontal triangle short; ocelli small; oc distinctly shorter than pvt (Figure 4C); antenna yellowish white; f_2_ extremely prolonged, much longer than f_3_ and ventrally with modified setosity (Figure 3C); M not reaching wing margin; A_1_ short (Figure 3H); male right cercus more slender than left cercus; posterior part of left gonostylus long and strongly curved; anterior lobe of right gonostylus very slender, with 2 setulae on apex (Figure 4A,B) ……...……… *C. clusioides* (Roháček)
-Frontal triangle long; ocelli large; oc as long as pvt (Figure 6B); antenna pale brown to ochreous-yellow; f_2_ as long as f_3_ (Figure 5B) and ventrally simply setulose; M reaching wing margin; A_1_ long (Figure 7D); male right cercus broad and short, left cercus slender and long; posterior part of left gonostylus long but less curved; anterior lobe of right gonostylus broad, subtriangular (Figure 6C) ……………………… *C. ornatus* sp. nov.Frons dark brown and coarsely striated along frontal triangle (Figure 20A); laterobasal sc almost as long as apical sc; wing membrane unicolorous dark brown (Figure 19F) …………………………...………………… *A. rugosus* sp. nov.
-Frons paler (light brown to yellowish orange) with smooth or only very finely transversely rugose surface (Figure 16B and Figure 28A); laterobasal sc 0.6–0.75 as long as apical sc; wing membrane variably coloured (sometimes as above) …………………….………………………………………… 41st flagellomere small, not longer or higher than pedicel (Figure 26B); anterior pa as long as sa or apical sc; 3 dc, middle as long as posterior dc (Figure 26F); wing broad (Figure 26I), with R_4+5_ strongly downcurved in basal half; CuA_1_ distinctly longer than dm-cu …………………………….………. *A. tertiaria* Hennig-1st flagellomere larger, longer and higher than pedicel; anterior pa distinctly longer than sa or apical sc (cf. Figure 16A and Figure 28B); only 2 dc, anterior (homologous with the middle dc of *A. tertiaria*) always distinctly shorter than posterior dc (Figure 16A); wing more elongate (Figure 17B and Figure 28G), with R_4+5_ less downcurved in basal half; CuA_1_ at most slightly longer than dm-cu ………………………………………………………… 5Thoracic scutum with dark brown and ochreous pattern (Figure 28B); f_2_ ventrally with single short distal row of six thicker setae; f_2_ distinctly but not strongly thicker than f_3_ (Figure 27A); wing darkened along anterior margin (Figure 28G) ………………………….…………. *A. willii* Pérez-de la Fuente, Hoffeins et Roháček
-Thoracic scutum more or less unicolorous, without distinct pattern (Figure 10C and Figure 14F); f_2_ ventrally simply finely setulose and markedly thicker than f_3_ (Figure 11B); wing membrane unicolorous pale or dark brown (Figure 12A and Figure 17A) ……………………………………………… 6Body dark blackish brown, with dark brown wings (Figure 10A); oc longer (Figure 11A); f_2_ strongly swollen and slightly longer than f_3_ (Figure 11B); t_2_ with a row of 10–12 relatively strong erect posterior setae (Figure 10D); t_3_ indistinctly bent (Figure 12F) ………………...……………….………………. *A. crassipes* sp. nov.
-Body pale ochreous to orange-ochreous, with pale ochreous-brown wings (Figure 14A); oc much shorter (Figure 16B); f_2_ strongly but less thickened and about as long as f_3_ (Figure 14G); t_2_ with a row of 8 or 9 relatively short erect posterior setae; t_3_ distinctly bent (Figure 16D) ……………………………………………………………… *A. luridus* sp. nov.


## 4. Discussion

The original placement of *Acartophthalmites* (albeit tentative) to the extant family Acartophthalmidae by Hennig [1] has already been recognized as incorrect by Roháček [20] because the species of the previous genus *Acartophthalmites* (now including all representatives of Clusiomitidae) are lacking all core characters (considered synapomorphies of Acartophthalmidae by [25,26]), viz. (1) eye distinctly haired, (2) C with humeral break, (3) no true vibrissa, (4) male sternites 6 and 8 fused to form a symmetrical pregenital synsclerite. Moreover, Acartophthalmidae also differ from Clusiomitidae (and Clusiidae) in having enlarged peristomal setae, widely spaced postvertical setae (being very close and strongly divergent in both Clusiomitidae and Clusiidae), an almost bare occiput, a markedly shortened arista, and 1–2 distinct dorsal preapical setae on f_3_. Buck [27]) even excluded Acartophthalmidae from Opomyzoidea (where it was placed together with Clusiidae in the suprafamily Clusioinea by [26]) and affiliated it under Carnoidea. Based on comparison of morphological characters, Roháček [20] and also Pérez-de la Fuente et al. [21] suggested that the fossil *Acartophthalmites* species in fact most closely resemble Clusiidae (including the Baltic amber fossil genus *Electroclusiodes* Hennig, 1965) and, particularly, the species *E. radiospinosa* Hennig, 1969 whose placement to Clusiidae and even to *Electroclusiodes* is sometimes doubted [28]. *Electroclusiodes radiospinosa*, although sharing with Clusiidae the synapomorphic structures of antennal pedicel with angular extension of outer distal margin and subcostal break, is also similar to *Clusiomitidae* in having t_2_ with a row of erect posterior (or posterodorsal) setae and R_1_ setulose, see [4] (Figure 24 and Figure 25). Already McAlpine [25,26] had previously suggested that not only *Electroclusiodes*, but also *Acartophthalmites tertiaria* Hennig, possibly belongs to Clusiidae.

### 4.1. Affiliation of the New Family

Undoubtedly, the new family Clusiomitidae is to be classified within the superfamily Opomyzoidea sensu McAlpine [26]. It shares almost all apomorphies defining this superfamily (except for internal postabdominal features invisible in fossils), including those considered by [26] (p. 1456) to be autapomorphic (AA) for this group, i.e., first flagellomere short, discoid; face membranous medially (it can be secondarily sclerotized); vibrissa present; wing contrastingly patterned (secondarily clear) (AA); C with subcostal break (secondarily entire, for situation in Clusiomitidae see below); cup short and convexly closed; A_1_ not attaining wing margin; stpl present; metasternal area bare; subscutellum well developed; male with sclerites of 7th abdominal segment reduced (T7 absent, S7 reduced and more or less fused with S8) (AA). Only the mesopleuron (anepisternum) has in Clusiomitidae the raised ridge along its upper posterior margin (considered AA for Opomyzoidea by [26]) distinctly reduced.

### 4.2. Relationships of Clusiomitidae

Within Opomyzoidea, the Clusiomitidae (and particularly species of the genus *Clusiomites*) externally most resembles Clusiidae. The latter family was thoroughly diagnosed by Lonsdale et al. [29]. However, Clusiomitidae do not share with Clusiidae the following putative synapomorphies of the latter family, cf. [29]: (1) dorsal subapical (dorsobasal in Clusiomitidae) arista, (2) triangular extensions on the outer and inner distal margins of the pedicel (at most slightly sinuate in Clusiomitidae), (3) a setose prosternum (bare in Clusiomitidae). These characters, missing in Clusiomitidae, demonstrate that this group cannot be affiliated under Clusiidae as currently delimited [29]. Clusiomitidae further differ from Clusiidae in having (1) short and strongly medially curved vi; (2) more than one row of postocular setulae; (3) apical sc never cruciate; (4) cx_3_ with posteroventral setose process; (5) t_1_ with 1 or 2 small dorsal preapical setae; (6) t_2_ with a row of erect posterior (or posterodorsal) setae and dorsally with one or two short setae in addition to one dorsal preapical; (7) t_3_ lacking dorsal preapical seta; (8) C entire, without subcostal break; (9) R_1_ with preapical kink at level of end of Sc; (10) male postabdomen with large dorsal pregenital sclerite (fusion of T6 + S8); (11) epandrium short, rather arch-shaped and ventrally open; (12) anal fissure of epandrium directed ventrally; (13) male cerci free, never fused basally. The majority of the above 13 characters are undoubtedly plesiomorphic but at least features no. 4, 7, 8, 9, 10 and 12 are treated as putative synapomorphies of Clusiomitidae which support its monophyly and validity. Their unusual t_2_ chaetotaxy (with a long row of posterior erect setae) is here considered a plesiomorphy shared with *Electroclusiodes radiospinosa* that was subsequently lost in Clusiidae. Moreover, the distinct kink on R_1_ (9) needs special comment. Such a kink is usually developed when Sc is terminally coalesced with R_1_ and C has a subcostal break (e.g., in Anthomyzidae [30]). However, in Clusiomitidae this kink (developed at the level of conjoinment of Sc with C) has probably a different origin. Most plausibly, the ancestor of Clusiomitidae had a subcostal break (as in Clusiidae) and the kink on R_1_ had been developed due to bending of the wings in this place; if this scenario is correct, then the subcostal break was lost subsequently and the entire C (character 8 above) is a synapomorphy of Clusiomitidae. There also seems to be a difference in the egg chorion sculpture: in Clusiidae the chorion is usually minutely (and variably) tuberculate with (usually) no more than one dozen longitudinal wrinkles [31] while in Clusiomitidae it is coarsely sculptured by irregular longitudinal rows of small, densely arranged and deep pits (see Figure 9D).

The only synapomorphies that appear to link Clusiomitidae and Clusiidae are the following three characters: (1) two pairs of postsutural dc, (2) a dorsal preapical seta on the t_2_, and (3) an apically bilobed and longitudinally divided female S8 (observed in *Clusiomites ornatus*, see Figure 8B). In addition, Clusiidae and Clusiomitidae share the following diagnostic characters (all plesiomorphic according to Lonsdale et al. [29]): (1) a complete subcosta ending far from apex of R_1_; (2) 1 vi; (3) oc divergent; (4) pvt divergent (but the latter much stronger in Clusiomitidae); (5) pvt closely-spaced and inserted at the inner margin of the posterior ocelli (treated as synapomorphic of Clusiidae in [26]); (6) 3 ors; (7) a soft and rather flat face. For these reasons (even though all putative synapomorphies are rather weak), we consider Clusiomitidae a probable sister group of Clusiidae, or, more precisely, of *Electroclusiodes radiospinosa* + Clusiidae.

### 4.3. Habitat Association and Diversity of Ancient Clusiomitidae

All known species of Clusiomitidae are known from Baltic amber fossils. They had surely lived in the so called “amber forest“. The Eocene subtropical to tropical amber forest developed in northern Europe (ranging from Fennoscandia to Ukraine) due to a very warm and humid climate in the Mid Eocene; the prevailing habitat was an oak and pine mountain forest [12]. According to [32] the plant community of the Baltic amber forest was represented by a thermophilous humid mixed forest with ecological parameters and the composition most similar to those of the modern subtropical plain forests of Eastern and Southeastern Asia. The putatively related Clusiidae are saproxylic flies associated with rotting wood in various types of (mainly humid) forests. Tschirnhaus & Hoffeins [13] recorded Clusiidae relatively frequently (compared to other Acalyptratae) in Baltic amber which indicates they were rather common inhabitants of the amber forest. This might also be true for Clusiomitidae, representatives of which are probably also saproxylic and confined to dead trees (syninclusions do not contradict this presupposition). In a mere 26 amber samples examined, at least 10 species of Clusiomitidae (seven described plus three or more unnamed) have been recognized. This fact indicates a relative high species diversity of the group in the Baltic amber forest.

## Data Availability

The data presented in this study are available in this article.

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
