# Peer review of "Clusiomitidae, A New Family of Eocene Fossil Acalyptratae, with Revision of Acartophthalmites Hennig and Clusiomites Gen. Nov. (Diptera)†"

_insects, 2021, doi:10.3390/insects12121123_

Round 1
Reviewer 1 Report
Acartophthalmites has long been a puzzling taxon to acalyptrate taxonomists, with most authors pointing out a general similarity to Clusiidae but stopping short of formally including Acartophthalmites in the Clusiidae. This paper is thus a tremendous contribution to the science, providing and exhaustively thorough description of the genus and a similar new genus, together with thorough descriptions of seven species. The illustrations are copious and exquisite, and the descriptions are meticulous and complete to a fault. This is a very important paper and warrants publication with very few corrections but, that said, I have a few differences of opinion with the authors. The more trivial suggestions are marked on the manuscript, but the most significant difference of opinion is the manner in which family status has been justified. Family status requires three main things: 1) Substantiation of monophyly (ie, definition), 2) practical means for recognition (ie, diagnosis), and 3) establishment that inclusion in a named family is not justified. Without major contributions to establishing these points it would be better to leave the status of Acartophthalmites (and its newly found apparent sister group) unchanged. The descriptive part of this paper exhaustively covers all features, without distinguishing putative synapomorphies from homoplasy and plesiomorphy. I'll therefore skip to the parts of the paper that justify family status for Clusiomitidae, and look at the three criteria:
1) Substantiation of monophyly: Both new taxa (genus and family) must be justified on the basis of apomorphic characters. Apomorphies are not clearly identified for the genera, although the key provides good diagnoses for the genera and setulose R1 does seem like a good synapomorphy for Acartophthalmites. More significantly, monophyly of the family is not addressed explicitly until part 4 (discussion). Here we see placement in Acartophthalmidae rejected, but that's not new, and we see comments about similarities between Clusiidae and the new family. Statements about synapomorphies between the new family and the superfamily Opomyzoidea are weak because Opomyzoidea is paraphyletic and the putative synapomorphies are hard to defend. But that is not a critical point – the important text falls under the heading "relationships of Clusiomitidae", in which we see that the new family "externally most resemble(s) the Clusiidae" but lacks many of the synapomorphies of extant Clusiidae; no putative synapomorphies are given linking Clusiomitidae to any other family. Thirteen characters are given distinguishing Clusiomitidae from Clusiidae, and six of them are then highlighted as synapomorphies of Clusiomitidae (the first explicit justification for the family in the paper!). Some are quite weak and strike me as plesiomorphic (t3 without dorsal preapical setae, C entire) and some (especially the lack of male T6) are intriguing and warrant comparison to other families, but they seem to justify the family as monophyletic. But the Clusiodinae is even more solidly justified as monophyletic and nobody is suggesting that it should be elevated to family status; monophyly it itself is not enough.
2) practical means for recognition (ie, diagnosis): this criterion is clearly met
3) establishment that inclusion in a named family is not justified: This is an important point, since three putative synapomorphies and several other similarities are identified linking Clusiomitidae to Clusiidae, and the authors conclude that Clusiidae (including Electroclusiodes, which has been treated as Clusiidae) is the sister group of Clusiidae. If that is really the case, what is the justification for treating it as a separate family rather than a basal lineage of Clusiidae? The Acalyptratae is already overloaded with small families of minimal predictive value, so the erection of a new family needs more explicit justification. This would still be a superb paper and a major contribution if all species were treated as Acartophthalmites (either unplaced or as a basal lineage in Clusiidae), but treatment of so few species in a new family and two genera demands a bit more justification. Even a single line at the end of the paper, explaining why it should not be left as an unplaced genus and why it would not be appropriate to treat it in the Clusiidae, would be useful.
All in all, this is a truly outstanding paper and I look forward to its publication. My different opinions are there for the authors to consider, but should not be considered to be significant criticism.

Author Response
All replyies to comments of the reviewer are in two files attached here (reviewed pdf MS, see answers in sticky notes + review). We would like to thank for all corrections and comments which contributed to enhance the text of the MS.

Reviewer 2 Report
This is a well-produced paper, carefully written and illustrated. The taxonomy of the studied taxa of Acalyptratae seems to be well supported. There appear to be not many issues with the text, neither with the drawings and photos. There are just a few format or language mistakes that the authors would easily notice, if they read the paper carefully again and follow my specific recommendations below.
Title
The title reads well but it sounds odd to say ‘revision of Clusiomites gen. nov.’. You can revise a genus that was established before your study but not right on your study.
Abstract
‘Spp. Nov.’ is not that correct, better place ‘sp. nov.’ After each new species name in the entire paper and every time you mention those species.
L23: ‘Established’ not ‘stablished’.
Introduction
The introduction is OK in general but in my opinion, it would be very useful to include some taxonomic information, for example, define morphologically the Acalyptratae or the genus Acartophthalmites as understood before your study. This is important to favour the understanding of the main taxa or Diptera groups you have as targets in your study, i.e. how they look like or how can they be separated from other groups.
L35-36: There is no need to mention the authority of a family, just mandatory for genera and species.
L53: ‘The present study is aimed at this revision’. Although it is rather clear what revision you are referring to in this line, it is better to say the revision of what when you start a new paragraph.
Material and methods
L68: ‘Amber specimens’ I would entitle this section as ‘Preparation of amber specimens’.
L85-86: I find a bit imprecise to say that you measured from the anterior margin of head, from which part specifically?
Results
L212: the first sentence in this line could be worded in other way for better understanding.
L249: scientific names written not in italics. Modify.
L376: maybe, better ‘conspicuously’ than ‘markedly’.
L428: the closing brackets are duplicated, delete one.
L473 and L485: there is a species name written not in italics. Modify.
L694: authors use here ‘spec. nov.’ but in other parts of the text ‘sp. nov.’. Please, always use the same notation.
L853: scientific names written not in italics. Correct. This mistake is in other parts of the text and authors must revise the entire document to find other cases.
Author Response
All replyies to comments of the reviewer are in the file attached here (review). We would like to thank for all corrections and comments which contributed to enhance the text of the MS.

Reviewer 3 Report
The present paper by Rohacek and Hoffeins is a fascinating exploration of a fossil group that appears to be well justified. Important features are exceptionally illustrated and described, making their erection of a new family and genus well justified and easily understood to the reader. The systematics sections are the most well-written, but the conclusions including the phylogenetic discussion are inconsistent and must be thoroughly reconsidered, and in some parts, entirely rewritten. There are many assumptions used, and there are departures from cladistic theory that make a number of points visibly ill-founded. Furthermore, the clade Opomyzoidea is also no longer accepted, and sections discussing the superfamily must be reassessed. It is also clear that unjustified and partially disproven assumptions made by previous authors are being carried over into this manuscript without being given adequate consideration or testing, resulting in visible bias affecting the present authors’ results.
Species are well-figured, clearly photographed and thoroughly described (but perhaps much too thoroughly, as I have mentioned to the senior author several times in the past for other manuscripts). The exceptionally verbose descriptions make it difficult to extract meaningful data, and the frequent inclusion of characters that are invariant between species of the genus or family needlessly clutter the descriptive text for individual species. The authors must remove all invariant characters and list them in the genus or family definition above.
Please refer to the edited PDF. Almost all comments on the manuscript are provided on that file.
There are some inconsistencies with respect to terminology, wording, text formatting and grammar, and there are a few minor mistakes, but these can be easily fixed. Some sections of text in the conclusions are long and the point being made is unclear, but again, these can also be remedied given minor effort. Clarity in the phylogenetic arguments will help this process significantly. Contradictions in key couplets should also be reassessed.

Author Response
All replyies to comments of the reviewer are in two files attached here (reviewed pdf MS, see answers in sticky notes + review). We would like to thank for all corrections and comments which contributed to enhance the text of the MS.
Because I am unable to submit both files, the pdf MS will be send directly to editor.

Round 2
Reviewer 1 Report
This is a superb paper, publishable .
I here attach the slightly annotated pdf . I offered some differences of opinion but they are subjective and for the author's interest only.

Author Response
All (minor) changes proposed by reviewer 1 are accepted and the MS corrected accordingly!
Reviewer 3 Report
The responses seem fine for the most part, although I still believe the phylogenetic components will not withstand basic basic scrutiny. Regardless, the methodology is transparent enough to allow reproduction and analysis by others, so it is scientifically acceptable.
Author Response
Because a more thourough relationships analysis is not possible at moment (as reviewer has accepted) we are glad to hear that reviewer 3 agrees it is acceptable in this form for publication. The revision of the English is unncessary because it was corrected at least by 2 English native dipterists (P. J. Chandler and S. A. Marshall).
J. Roháček